# Pre-trained Hypergraph Convolutional Neural Networks with Self-supervised Learning

**Yihe Deng**[*]  
*Department of Computer Science*  
*University of California, Los Angeles*

*yihedeng@cs.ucla.edu*

**Ruochi Zhang**[*]  
*Eric and Wendy Schmidt Center*  
*Broad Institute of MIT and Harvard*

*zhangruo@broadinstitute.org*

**Pan Xu**  
*Department of Biostatistics and Bioinformatics*  
*Duke University*

*pan.xu@duke.edu*

**Jian Ma**[†]  
*Ray and Stephanie Lane Computational Biology Department*  
*School of Computer Science*  
*Carnegie Mellon University*

*jianma@cs.cmu.edu*

**Quanquan Gu**[†]  
*Department of Computer Science*  
*University of California, Los Angeles*

*qgu@cs.ucla.edu*

**Reviewed on OpenReview:** *https://openreview.net/forum?id=OVWXWPmctm*

## Abstract

Hypergraphs are powerful tools for modeling complex interactions across various domains, including biomedicine. However, learning meaningful node representations from hypergraphs remains a challenge. Existing supervised methods often lack generalizability, thereby limiting their real-world applications. We propose a new method, Pre-trained Hypergraph Convolutional Neural Networks with Self-supervised Learning (PhyGCN), which leverages hypergraph structure for self-supervision to enhance node representations. PhyGCN introduces a unique training strategy that integrates variable hyperedge sizes with self-supervised learning, enabling improved generalization to unseen data. Applications on multi-way chromatin interactions and polypharmacy side-effects demonstrate the effectiveness of PhyGCN. As a generic framework for high-order interaction datasets with abundant unlabeled data, PhyGCN holds strong potential for enhancing hypergraph node representations across various domains.

## 1 Introduction

Hypergraphs are essential data structures adept at modeling complex multi-entity relationships. Conventional graph neural networks (GNNs), limited to pairwise interactions, cannot effectively capture the higher-order interactions inherent to hypergraphs. This shortfall led to previous methods (Agarwal et al., 2005; 2006; Satchidanand et al., 2015) expanding hypergraphs to graphs using clique expansion (Sun et al., 2008). More recent works (Yadati et al., 2019; Dong et al., 2020; Arya et al., 2020; Bai et al., 2021; Yi & Park, 2020; Sun et al., 2021) have adopted GNN structure such as graph convolutional networks (GCNs) and

---

[*]Equal Contribution  
[†]Co-corresponding Authors

extended them to the hypergraph setting. HyperGCN (Yadati et al., 2019) and HNHN (Dong et al., 2020), in particular, achieved state-of-the-art performances on hypergraph benchmark datasets. However, these supervised methods heavily rely on labeled nodes, which are often scarce in practical scenarios. In situations with insufficient labels, these methods fail to fully utilize the rich structural information of hypergraphs, leading to less effective and poorly generalizable node representations.

Self-supervised learning (SSL), an approach that extract meaningful knowledge from abundant unlabeled data to enhance model generalization (Doersch et al., 2015; Kolesnikov et al., 2019), presents a promising strategy to address these challenges. SSL creates pretext tasks from unlabeled data to predict unobserved input based on the observed part, enhancing the generalizability of models trained for computer vision (e.g., ImageNet pre-training (Girshick et al., 2014; He et al., 2019)) and natural languages (e.g., BERT (Devlin et al., 2019)). While SSL methods have been proposed for GNNs (Hu et al., 2020a; You et al., 2020b; Hu et al., 2020b; Wu et al., 2021; Jin et al., 2022; Hwang et al., 2020; Hao et al., 2021; Sun et al., 2020), their application to hypergraph learning is underexplored. Noteworthy works (Xia et al., 2021; Yu et al., 2021; Xia et al., 2022) have targeted specific applications such as recommender systems, while others (Du et al., 2021) propose pre-training GNNs on hypergraphs, primarily focusing on hyperedge prediction. A detailed discussion of related works can be found in Appendix A.

In this work, we introduce PhyGCN to address the urgent need for a general method that can effectively leverage unlabeled data from hypergraphs to enhance node representation learning. This self-supervised method extracts knowledge from the hypergraph structure using self-supervised tasks, generating robust node representations for diverse downstream tasks. Constructing self-supervised tasks in hypergraphs poses a unique challenge due to the variable sizes of hyperedges. We tackle this challenge by designing a self-supervised task that predicts masked hyperedges from observed ones and by incorporating an attention mechanism into our model architecture (Zhang et al., 2020) to predict variable-sized hyperedges, which allows the size of the hyperedge to vary (e.g., the size $k$ can arbitrarily be 2, 3, or 4). In contrast, fixed-sized hyperedge prediction involves predicting hyperedges that have a single, constant size.

Link prediction, an effective pre-training task for traditional graphs, typically involves designing a neural network with fixed-length input to predict edges. Hyperedge prediction, however, is more challenging due to variable sizes. Simple average pooling of node embeddings has been found insufficient to model a hyperedge (Tu et al., 2018). Although some works (Xia et al., 2021; Yu et al., 2021; Xia et al., 2022; Du et al., 2021) have explored the potential of self-supervised learning on hypergraphs, our PhyGCN method is the first straightforward and effective approach to utilize hypergraph structure and can be generally applied to many tasks.

We demonstrate the effectiveness of PhyGCN through various evaluations across multiple tasks and datasets, showing its advantage over state-of-the-art hypergraph learning methods. Notably, PhyGCN has been applied to study multi-way chromatin interactions and polypharmacy side-effect network data, confirming its advantages in producing enhanced node embeddings and modeling higher-order interactions. Together, PhyGCN, a self-supervised method for hypergraph representation learning, can be applied broadly to a variety of problems.

## 1.1 Most Related Work

In this subsection, we present the most related work to ours that focus on hypergraph pretraining/self-supervised learning. Self-supervised learning and pre-training for hypergraphs remain largely unexplored. The majority of works (Xia et al., 2021; Yu et al., 2021; Xia et al., 2022) focused on specific applications in recommender systems. Xia et al. (2021) introduced Dual Channel Hypergraph Convolutional Networks (DHCN), explicitly designed for session-based recommendation. DHCN's self-supervised learning task seeks to maximize mutual information between session representations learned by a graph convolutional network and DHCN, aiming to enhance hypergraph network learning. Yu et al. (2021) also focused on recommender systems, leveraging higher-order interactions to improve recommendation quality. They introduced a self-supervised task to compensate for the loss incurred by their multi-channel hypergraph convolutional network, aiming to maximize mutual information concerning the representations of the user node, the neighboring sub-hypergraph of the user, and the entire hypergraph. Xia et al. (2022) proposed to enhance recommendation and user representations via a hypergraph transformer network that uses a cross-view generative

self-supervised learning for data augmentation. Lastly, Du et al. (2021) proposed pre-training with GNNs on hypergraphs for hyperedge classification. This method uses GNNs with self-supervised pre-training tasks, achieving state-of-the-art performances on both inductive and transductive hyperedge classification. Behrouz et al. (2024) studied temporal hypergraphs and used random walk to automatically extract temporal, higher-order motifs. Telyatnikov et al. (2023) studied how the concept of homophily can be characterized in hypergraphs, and proposed a novel definition of homophily to effectively describe HNN model performances. Both works are related to our study on hypergraphs but orthogonal to our focus on the performance improvement by self-supervised learning on hypergraphs. Lee & Shin (2023) studied self-supervised learning on hypergraphs and utilized a tri-directional contrastive loss function consisting of node-, hyperedge-, and membership-level contrast. Very recently, Kim et al. (2024) studied generative self-supervised learning on hypergraphs and proposed a new task called hyperedge filling for hypergraph representation learning. Lastly, we take an additional note on Hyper-SAGNN (Zhang et al., 2020), which provides the attention mechanism used in PhyGCN. HyperSAGNN is a method designed for hyperedge prediction, and is therefore only considered as a baseline on hyperedge prediction tasks. For node classification tasks and other tasks we focus on, comparing with HyperSAGNN is not possible.

## 2 Preliminaries

### 2.1 Graph Neural Networks

We denote a graph by $G = (V, E)$, where $V$ is the vertex set with node $v \in V$ and $E$ is the edge set with pairwise edge $(u, v) \in E$. Graph neural networks (GNNs) mostly follow the scheme (Xu et al., 2019): $\mathbf{x}_v^l = \text{COMBINE}^{(l)}\big(\mathbf{x}_v^{(l-1)}, \text{AGG}^{(l)}\big(\{\mathbf{x}_u^{(l-1)} : u \in \mathcal{N}(v)\}\big)\big)$, where $\mathbf{x}_v^{(l)}$ is the embedding vector of node $v$ at layer $l$ of the network, $\mathbf{x}_v^{(0)}$ is the input data, and $\mathcal{N}(v)$ is the set of nodes that are adjacent to $v$. Here $\text{AGG}(\cdot)$ and $\text{COMBINE}(\cdot)$ are two functions usually used to aggregate the neighboring embeddings and combine the aggregated embedding with the original node embedding to output the embedding of a given node. Different GNN models may have different choices of $\text{COMBINE}(\cdot)$ and $\text{AGG}(\cdot)$ functions. As a most popular approach to aggregate and combine the node vectors, GCN (Kipf & Welling, 2017) averages over all the first-order neighboring nodes including the node itself. Let $\mathbf{X}^{(l)}$ represent the embedding matrix of all nodes at layer $l$, and the equation boils down to $\mathbf{X}^{(l)} = \sigma\big(\widehat{\mathbf{A}}\mathbf{X}^{(l-1)}\mathbf{\Theta}^{(l-1)}\big)$, where $\sigma$ is some activation function, $\widehat{\mathbf{A}} \in \mathbb{R}^{N \times N}$ is the normalized adjacency matrix with N denoting the number of nodes. $\mathbf{\Theta} \in \mathbb{R}^{h_{l-1} \times h_l}$ is the weight matrix at layer $l-1$ where $h_l$ as the hidden embedding size at each layer $l$. And $\mathbf{X}^{(0)} \in \mathbb{R}^{N \times d_0}$ is the input node features of all nodes where $d_0$ is the size of the input node feature size.

### 2.2 Hypergraph Convolutional Networks

In an attempt to adapt GCN for tasks on hypergraph, the problem naturally arises: how do we appropriately adapt the adjacency matrix for hypergraphs? Adjacency matrix is natural to construct for graphs: setting the entry $\mathbf{A}_{ij} = 1$ for each edge $(v_i, v_j)$. However, hyperedge involve more than two nodes and constructing $\mathbf{A}$ will thus be tricky. As Feng et al. (2019) proposed, for a hypergraph $G = (V, E)$, a hypergraph can be represented by an incidence matrix $\mathbf{H} \in \mathbb{R}^{N \times M}$, with $N$ as the number of nodes and $M$ as the number of hyperedges. When a hyperedge $e_j \in E$ is incident with a vertex $v_i \in \mathcal{V}$, $\mathbf{H}_{ij} = 1$, otherwise equals 0. We denote $\mathbf{W}$ as the hyperedge weight matrix, and then a straightforward definition for the adjacency matrix will be $\mathbf{A} = \mathbf{HWH}^\top$. And to normalize it similarly as in GCN, we take $\widehat{\mathbf{A}} = \mathbf{D}_v^{-1/2}\mathbf{HWD}_e^{-1}\mathbf{H}^\top\mathbf{D}_v^{-1/2}$, where the diagonal matrices $\mathbf{D}_e \in \mathbb{R}^{M \times M}, \mathbf{D}_v \in \mathbb{R}^{N \times N}$ respectively represent the degree of hyperedges and the degree of vertices (Feng et al., 2019) and $\widehat{\mathbf{A}} \in \mathbb{R}^{N \times N}$. A layer of such hypergraph GCN model will then be similar to the regular GCN but with a different adjacency matrix $\mathbf{X}^{(l)} = \sigma\big(\widehat{\mathbf{A}}\mathbf{X}^{(l-1)}\mathbf{\Theta}^{(l-1)}\big)$. Similarly, $\mathbf{\Theta} \in \mathbb{R}^{h_{l-1} \times h_l}$ is the weight matrix at layer $l-1$ and $\mathbf{X}^{(0)} \in \mathbb{R}^{N \times d_0}$ is the input node features of all nodes. We adopt such simple architectures in our work, along with several techniques in GNNs to enhance the model's learning on both self-supervised and main tasks.

## 3 Our Proposed Method

In this section, we present our framework of learning from Pre-trained Hypergraph Convolutional Neural Networks with Self-supervised Learning (PhyGCN). We start with an overview of the workflow of our

framework in Section 3.1. We then present the detailed network structure design and the self-supervised pre-training pipeline in Sections 3.2 and 3.3 respectively.

## 3.1 Overview of PhyGCN

Figure 1: Overview of PhyGCN. **a.** The general workflow of PhyGCN begins with a hypergraph and its hyperedge information. Hyperedge prediction is employed as a self-supervised task for model pre-training. The pre-trained embedding model is then used for downstream tasks. **b.** This illustration elaborates on PhyGCN's detailed workflow, which involves generating labeled data for self-supervised learning by randomly taking 80% of the hyperedges as positive samples, and creating negative samples by changing the nodes in each positive sample. This data facilitates the training of the base hypergraph convolutional network with the attention network. The pre-trained base model is subsequently used for downstream tasks with a fully connected layer.

**Fig. 1** provides a detailed depiction of PhyGCN's model architecture and general workflow. PhyGCN aims to enhance node representation learning in hypergraphs by effectively leveraging abundant unlabeled data. To this end, we propose 1) a self-supervised learning strategy that pre-trains the model directly on the hypergraph structure before proceeding with any downstream tasks, and 2) a corresponding model architecture comprising a base hypergraph convolutional network for representation learning, and an attention network (Zhang et al., 2020) for executing the self-supervised task.

As shown in **Fig. 1a**, to pre-train the model, we establish a self-supervised hyperedge prediction task using randomly masked hyperedges (20%). The neural networks are then trained to reconstruct them using only the remaining hyperedges (80%). We emphasize that, the pre-training is done on the available *hyperedges* with negative samplings to construct the negative examples (Section 3.3 provides the formal definition). The pre-trained neural network can then be used for downstream tasks such as node classification and

continual signal prediction. See Section 3.3 for the details of the pre-training scheme. **Fig. 1b** details the architecture of the neural network model, consistent of a base hypergraph convolutional network and two end networks for different tasks. The base convolutional network computes the embedding of each node in the hypergraph, which is then directed either to the attention network for variable-sized hyperedge prediction, or to a fully connected layer for node-level downstream tasks. During the pre-training phase, the base model and the attention network are trained while the fully connected layer for downstream tasks remains randomly initialized. We select and retain the base model that exhibits the best performance on the pre-training task (hyperedge prediction) on the validation set, and subsequently fine-tune the base model together with the fully connected layer for downstream tasks. During this stage, we incorporate the original hypergraph data with its complete hyperedge information, and use the downstream task labels as the supervisory signals for training.

To calculate the embedding for a target node, the hypergraph convolutional network aggregates information from neighboring nodes connected to it via hyperedges, and combines it with the target node embedding to output a final embedding. These computations are carried out at each layer of the network. The base hypergraph convolutional network is a stack of such hypergraph convolutional layers (see Section 3.2). To mitigate over-smoothing (Chen et al., 2020) and enhance the performance of the base model, we concatenate the output of each layer in the end to generate the final embeddings for the nodes. Furthermore, we adapt DropEdge (Rong et al., 2020) and introduce DropHyperedge, where we randomly mask the values of the adjacency matrix for the base hypergraph convolutional network during each training iteration to prevent overfitting and improve generalization in the pre-training phase. By integrating the model architecture and pre-training scheme, PhyGCN effectively learns from a hypergraph structure and applies the learned knowledge to downstream tasks.

## 3.2 Base Hypergraph Convolutional Network

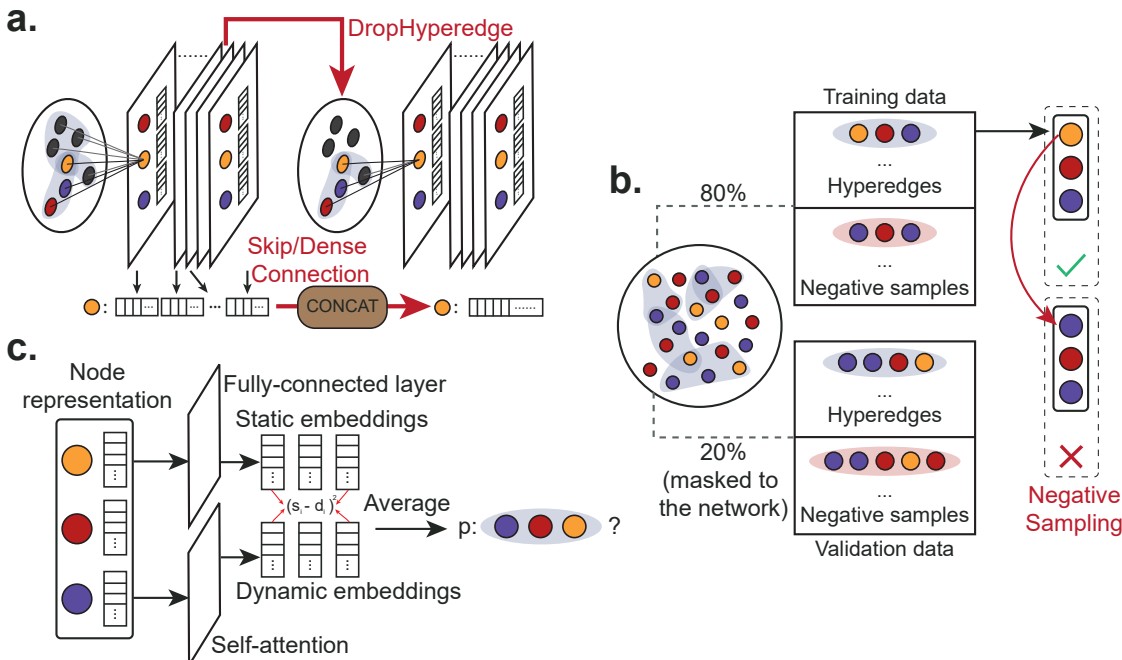

Figure 2: Illustration of the method. **a.** For the base hypergraph convolutional network, we propose two adapted strategies (DropHyperedge and Skip/Dense Connection) to further improve the pre-training of PhyGCN. **b.** For the self-supervised pre-training task, we randomly mask 20% of the hyperedges in the hypergraph and generate negative samples with regard to each positive training example. See Section 3.3 for more details. **c.** The pre-training scheme requires prediction of hyperedges that are arbitrarily sized. Therefore, we adopt an attention network (Zhang et al., 2020) to fulfill the task.

We thus propose a self-supervised learning strategy, along with a simple but effective hypergraph convolutional network architecture, to better capture the structural knowledge of a hypergraph. Our base convolutional model follows from the previous setup of a hypergraph convolutional network, where $\mathbf{X}^{(l)} = \sigma\big(\widehat{\mathbf{A}}\mathbf{X}^{(l-1)}\mathbf{\Theta}^{(l-1)}\big)$. And we further introduce the following modifications to the base hypergraph convolutional model.

**Skip/Dense Connection.** GCNs have been known to the have vanishing gradient problem where stacking multiple convolution layers in a GCN model causes the derivative of the loss function to approach zero and thus hard to train (Wu et al., 2020). To make GCNs deeper and prevent the vanishing gradient problem, Li et al. (2019) proposes several adapted strategies based on the methods frequently used for CNNs. Similar vanishing gradient problems have also been observed in hypergraph convolutional networks (Chen & Zhang, 2022). Therefore, we design a similar but more computing-efficient strategy where we concatenate the outputs of all layers of the model only after the last layer (instead of concatenating the previous outputs at *each* layer as in Li et al. (2019)), as illustrated in Figure 2a. That is, we keep all the previous layers unchanged and set the last layer of the convolutional model as:

$$\boldsymbol{X}^{(n)} = \big[\sigma(\widehat{\boldsymbol{A}}\boldsymbol{X}^{(n-1)}\boldsymbol{\Theta}^{(n-1)}), \boldsymbol{X}^{(n-1)}, ..., \boldsymbol{X}^{(0)}\big]. \tag{1}$$

And we denote $\mathbf{x}_j^{(l)}$ as the $j$-th row of $\mathbf{X}^{(l)}$, which is the embedding at layer $l$ for node $j$. Such design helps us save computation time while preventing the model's performance from dropping with more layers. The eventual base model we use is then the $n$-layer convolutional model with concatenation at the last layer. We either add an attention layer for the self-supervised task or an MLP layer for the main task.

**DropHyperedge.** Overfitting is a common issue in the use of neural networks, where the network learns to fit the training data too well and fails to generalize to test data. To mitigate such an effect, dropout has been a popular technique for many fields and is also commonly used for GCNs. To further reduce overfitting and improve generalization in GCNs, Rong et al. (2020) introduced a new regularization technique designed for GCNs called DropEdge, which randomly mask the edges of the graph at each iteration. In this work, to address the over-fitting issues with PhyGCN and encourage better generalization in the pre-training stage, we adapt the DropEdge strategy to our model by randomly masking the values on the adjacency matrix $\widehat{\mathbf{A}}$ at each iteration (as illustrated in Figure 2a). Note that such a strategy is only used for the self-supervised hyperedge prediction task. We use the same dropout strategy in the main task as the other baselines.

### 3.3 Pre-training with Self-supervisory Signals

**Self-supervised task.** As shown in Figure 2b, given a hypergraph $G = (V, E)$, we randomly divide the set of hyperedges $E$ into the training set $E_{\text{train}}$ and the validation set $E_{\text{valid}}$. Moreover, we generate 5 negative samples for each hyperedge in $E_{\text{train}}$ and $E_{\text{valid}}$, and denote the negative sample sets as $E'_{\text{valid}}$ and $E'_{\text{valid}}$. To distinguish these generated samples, we refer to $E_{\text{train}}$ and $E_{\text{valid}}$ as the positive training set and the positive validation set respectively. Combining these datasets, we obtain the final training and validation set as follows.

$$S_{\text{train}} = \big\{(E_i, z_i) \mid E_i \in E_{\text{train}}, \text{ for } z_i = 1; E_i \in E'_{\text{train}}, \text{ for } z_i = 0 \big\},$$

and

$$S_{\text{valid}} = \big\{(E_i, z_i) \mid E_i \in E_{\text{valid}}, \text{ for } z_i = 1; E_i \in E'_{\text{valid}}, \text{ for } z_i = 0 \big\},$$

where $E_i = \{v_1, \dots, v_k\}$ is a set of nodes of arbitrary size $k$. We then train the model by the hyperedge prediction task: given a set of node features $\{\mathbf{x}_j^{(0)}\}_{j \in [E_i]}$ with arbitrary size $k$ and $\mathbf{x}_j^{(0)} \in \mathbb{R}^{d_0}$, the goal is to predict $z_i \in \{0, 1\}$: whether the nodes form a hyperedge. Therefore, given the training set $S_{\text{train}}$, we aim to minimize the following cross-entropy loss with regard to each training batch $S$:

$$\mathcal{L}_{\text{pretrain}} = -\tfrac{1}{|S|} \sum_{i \in [S], S \in S_{\text{train}}} z_i \cdot \log\big(g\big(\{f(\mathbf{x}_j^{(0)})\}_{j \in [E_i]}\big)\big). \tag{2}$$

Here, $f(\cdot) : \mathbb{R}^{d_0} \to \mathbb{R}^{d_1}$ denotes the base hypergraph convolution model, where $d_0$ is the input feature size and $d_1$ is the size of the output embedding $\mathbf{x}_j^{(n)}$ of the network. Moreover, $g(\cdot) : \big\{\mathbb{R}^{d_1}\big\}_{|E_i|} \to [0, 1]$ denotes the attention network, which takes a arbitrary-size set of node embeddings $\mathbf{x}_j^{(n)}$ and output the probability $p$ of whether the nodes form a hyperedge. The base hypergraph convolutional model $f(\cdot)$ and attention

network $g(\cdot)$ are trained together, and the base model's weight parameters are saved for further fine-tuning in downstream tasks.

**Negative sampling.** We generate negative samples similarly to Zhang et al. (2020); Tu et al. (2018). Specifically, for each hyperedge $E_i = \{v_1, \ldots, v_k\} \in E_{\text{train}}$, which is referred to as the positive sample, we generate 5 negative samples. Each negative sample $E_i'$ is obtained by randomly alternating one or multiple of the nodes in $E_i$ such that $E_i'$ does not belong to the existing hypergedge set $E_{\text{train}}$ or $E_{\text{valid}}$, as shown in Figure 2b. Moreover, we vary the sampling strategy for different hypergraphs based on their node types. For hypergraphs with only one node type (e.g., citation networks), we have $E_i' = \{v_1, \ldots, v_j', \ldots, v_k\}$ : $E_i' \notin \cup\{E_{\text{train}}, E_{\text{valid}}\}$. For hypergraphs with multiple node types (e.g., polypharmacy side-effect data), we randomly alter a node within either node type. Take a hypergraph with two node types $v$ and $u$ for example. The positive hyperedge takes the form of $E_i = \{v_1, \ldots, v_k, u_1, \ldots, u_l\}$. The negative samples are then designed as

$$E_i' = \{v_1, \ldots, v_j', \ldots, v_k, u_1, \ldots, u_l\} : E_i' \notin \cup\{E_{\text{train}}, E_{\text{valid}}\},$$
$$E_i' = \{v_1, \ldots, v_k, u_1, \ldots, u_j', \ldots, u_l\} : E_i' \notin \cup\{E_{\text{train}}, E_{\text{valid}}\}.$$

Finally, for hypergraphs with fixed-length hyperedges, such as the polypharmacy dataset (Zitnik et al., 2018), we propose an enhanced negative sampling strategy. Assume the fixed length is $k = 3$ and the positive sample is $E_i = \{v_1, v_2, v_3\}$. We generate the negative sample $E_i' = \{v_1', v_2', v_3\}$ by altering two nodes such that (1) $E_i' \notin \cup\{E_{\text{train}}, E_{\text{valid}}\}$ and (2) there exists a positive sample $E_j = \{v_1', v_2', v_3'\} \in \cup\{E_{\text{train}}, E_{\text{valid}}\}$. This ensures that the altered nodes in the negative sample interact in another positive sample.

Our enhanced negative sampling strategy is novel in self-supervised learning and provides more accurate prediction in differentiating positive and negative samples in hypergraph applications. For example, in the polypharmacy side effect dataset, where we aim to predict whether a drug combination causes a specific side effect, traditional approaches (Zitnik et al., 2018; Nováček & Mohamed, 2020) might generate random drug entities that do not cause any side effect. In contrast, our enhanced negative sampling strategy generates drug entities that cause other side effects, forcing the models to learn more subtle differences between the positive and negative hyperedges. See Section 4.3 and Figure 5 for more details.

**Attention Network.** As in Figure 2c, the attention network follows the design in Zhang et al. (2020) with weight matrices $\mathbf{W}_Q \in \mathbb{R}^{d_1 \times d_\alpha}$, $\mathbf{W}_K \in \mathbb{R}^{d_1 \times d_\alpha}$, and $\mathbf{W}_V \in \mathbb{R}^{d_1 \times d_2}$. Given the a node set $E$ of size k and the node embeddings $\{\mathbf{x}_j^{(n)} = f(\mathbf{x}_j^{(0)})\}_{v_j \in E}$, the normalized attention coefficient $\alpha_{il}$ regarding node $v_i$ and node $v_l$ is computed as $\alpha_{il} = \exp\left((\mathbf{W}_Q^\top \mathbf{x}_i^{(n)})^\top (\mathbf{W}_K^\top \mathbf{x}_l^{(n)})\right) / \sum_{j=1}^{k} \exp\left((\mathbf{W}_Q^\top \mathbf{x}_i^{(n)})^\top (\mathbf{W}_K^\top \mathbf{x}_j^{(n)})\right)$. Then, the dynamic embedding of each node $v_i \in E$ is computed as $\mathbf{d}_i = \tanh\left(\sum_{l=1}^{k} \alpha_{il} \mathbf{W}_V^\top \mathbf{x}_l\right) \in \mathbb{R}^{d_2}$. And with weight matrix $\mathbf{W}_S \in \mathbb{R}^{d_1 \times d_2}$, the static embedding of each node $v_i \in E$ is computed as $\mathbf{s}_i = \mathbf{W}_S^\top \mathbf{x}_i \in \mathbb{R}^{d_2}$. Finally, with another layer $\mathbf{W}_O \in \mathbb{R}^{d_2}$, the output of the attention network is then the probability that the node set is a hyperedge $p = 1/k \sum_{i=0}^{k} \sigma\left(\mathbf{W}_O^\top (\mathbf{d}_i - \mathbf{s}_i)^{\circ 2} + b\right)$. where $\sigma$ is the sigmoid function and $b \in \mathbb{R}$. Notation $\circ$ denotes element-wise power operation.

## 4 Experiments

Based on the pre-trained model $f(\cdot)$ in Section 3 that generates node embeddings with the information it leveraged from the pre-training task, we conduct various experiments on node-level downstream tasks including classification problems on citation networks, biological discoveries in multi-way chromatin interaction datasets, and predicting polypharmacy side effects. In particular, we load the saved base model and add a fully connected layer for prediction. Then we fine-tune the model by the downstream task.

Let $h(\cdot)$ denote the fully connected layer that is task-specific for different data. For node classification tasks with $C$ classes as in Section 4.1, we have $h(\cdot) : \mathbb{R}^{d_1} \to \mathbb{R}^C$. Given the training set $S_{\text{train}}$ with input feature $\mathbf{x}_i^{(0)} \in \mathbb{R}^{d_0}$ for each node $i$ the label $y_i \in [C]$ representing the node class, we minimize the following cross-entropy loss with regard to each training batch $S$:

$$\mathcal{L}_{main} = -\frac{1}{|S|} \sum_{i \in [S], S \in S_{\text{train}}} \log\left(\exp\left(h\left(f(\mathbf{x}_i^{(0)})\right)_{y_i}\right) / \exp\left(\sum_{c=1}^{C} h\left(f(\mathbf{x}_i^{(0)})\right)_c\right)\right). \tag{3}$$

For continual signal prediction tasks as in Section 4.2, we have $h(\cdot): \mathbb{R}^{d_1} \to \mathbb{R}$. Given the training set $S_{\text{train}}$ with input feature $\mathbf{x}_i^{(0)} \in \mathbb{R}^{d_0}$ for each node $i$ the label $y_i \in \mathbb{R}$, we minimize the following mean squared error with regard to each training batch $S$:

$$\mathcal{L}_{main} = 1/|S| \sum_{i \in [S], S \in S_{\text{train}}} (h(f(\mathbf{x}_i^{(0)})) - y_i)^2. \tag{4}$$

In the downstream tasks, we keep fine-tuning the base convolutional network $f(\cdot)$ with the task-specific fully connected layer $h(\cdot)$.

## 4.1 Self-supervised learning improves performance

We start with the fundamental question: *can PhyGCN more effectively utilize the structural information in a hypergraph?* To assess this, we benchmarked PhyGCN against the current state-of-the-art hypergraph learning methods, including HyperGCN (Yadati et al., 2019) and HNHN (Dong et al., 2020). We also established a baseline by merely decomposing the hypergraphs into graphs and applying a standard GCN (Kipf & Welling, 2017). We evaluated these methods using the benchmark node classification task on citation networks (including Citeseer, Cora, DBLP, and PubMed) whose hyperedges denote either co-authorship or co-citation relationships. We note that, citation networks are modeled as hypergraphs where hyperedges contain publications sharing the same citation (co-citation) or the same author (co-authorship). Since multiple publications can cite the same paper or share the same author, they form hyperedges with arbitrary sizes (i.e., variable-sized hyperedge). Given that the baselines utilize different data splits for their evaluations, we consistently evaluated our method along with all the baselines on the same random data splits across a range of training data ratios. Detailed statistics, experiment settings, and results are provided in Appendix B.2.

In order to ascertain how each hypergraph learning method performs with limited training data, we first set the training data ratio to 0.5% across the five datasets. The results are shown in **Fig. 3a**. Across 10 random data splits, we found that both HyperGCN and HNHN fail to outperform the standard GCN model, which merely transforms the hypergraphs into conventional graphs. Conversely, PhyGCN, equipped with pre-training on the self-supervisory signals obtained directly from the hypergraph, performs significantly better when dealing with a small training data ratio. It outperforms GCN across all datasets, particularly on small networks such as Citeseer and Cora. The improvement over GCN suggests that PhyGCN in more adept at learning from the higher-order hypergraph structure as opposed to merely decomposing it into a binary graph. Furthermore, the pre-training scheme allows PhyGCN to extract more information for node representation learning, especially when dealing with small hypergraphs.

To evaluate PhyGCN against the baselines on a wider range of training data ratios, we generated data splits with increasing training data ratios of 1%, 2%, 4%, 8% and 16%. The results on the Cora (co-authorship) dataset are shown in **Fig. 3b**, while comprehensive results of all models on the five datasets are provided in Appendix B.2. Overall, PhyGCN outperforms the other methods, demonstrating more robust performance under random data splits. Across varying training data ratios, PhyGCN consistently outperforms the three baselines. At training data ratios of 2%, 4%, 8% and 16%, PhyGCN's performance is considerably more stable than that of the baselines, as indicated by the significant decrease in the variance of PhyGCN's accuracy across 10 random data splits.

These observations suggest that the pre-training stage enables the model to more effectively gather information from unlabeled data, allowing it to perform well even when a small amount of data is available. The results underscore the potential of PhyGCN in learning with higher-order interactions without the strong requirement of ample labeled data for a specific task. By fully leveraging the hypergraph structure for self-supervised training and designing a simple but effective model architecture, PhyGCN can achieve superior and more stable performances on downstream tasks compared to existing baselines.

**Datasets.** The details of the datasets used for the evaluation of node classification are shown in **Table 1**. It is noteworthy that the data used by Yadati et al. (2019) contains a lot of isolated nodes, implying that a significant portion of nodes, not part of any hyperedge, are classified solely based on their feature information. This is not an ideal setting for evaluating the capacity of a graph/hypergraph neural network in capturing structural information. We opt for a similar setting to Dong et al. (2020), where we remove the isolated nodes from the raw data. Moreover, since the compared works Yadati et al. (2019); Dong et al.

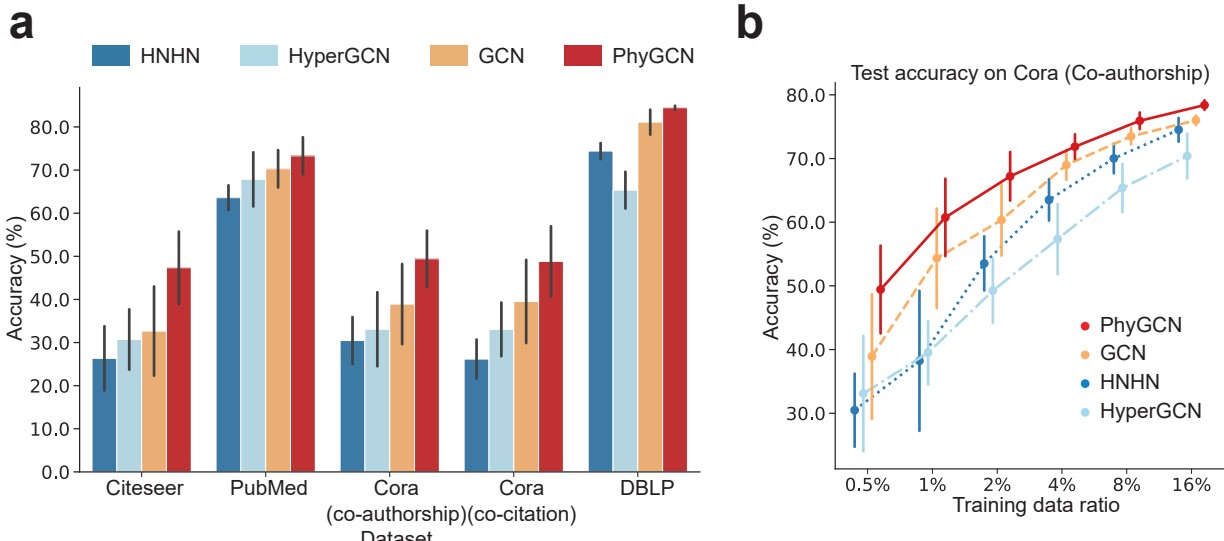

Figure 3: Evaluation of our model on node classification on citation networks with variable hyperedge sizes (Citeseer, Cora (co-authorship), Cora (co-citation), DBLP, and PubMed). The baseline methods include HNHN (Dong et al., 2020), HyperGCN (Yadati et al., 2019), and vanilla GCN (Kipf & Welling, 2017). **a.** Comparison of PhyGCN's accuracy with the baseline methods when the training data is minimal (0.05%). **b.** On the Cora (co-citation) dataset, PhyGCN consistently outperforms the baselines across various training data ratios (0.05%, 1%, 2%, 4%, 8%, 16%).

(2020) use different train-test ratios for each dataset, we make a uniform setting of ratios for all data and models: 0.5%, 1%, 2%, 4%, 8% and 16%. Specifically, we conduct four experiments on each dataset for each model, incrementally increasing the training data ratio from 0.5% to 16%. Training data with a larger proportion includes the nodes from the training data of a smaller proportion. Lastly, we perform 10 random splits for each ratio of each dataset to evaluate the average performance alongside the standard deviation.

|  | DBLP (co-authorship) | Pubmed (co-citation) | Cora (co-authorship) | Cora (co-citation) | Citeseer (co-citation) |
|---|---|---|---|---|---|
| # nodes, $|V|$ | 41302 | 3840 | 2388 | 1434 | 1458 |
| # hyperedges, $|E|$ | 22363 | 7963 | 1072 | 1579 | 1079 |
| avg hyperedge size | $4.5 \pm 5.4$ | $4.3 \pm 5.7$ | $4.3 \pm 4.2$ | $3.0 \pm 1.0$ | $3.2 \pm 2.0$ |
| # features, d | 1425 | 500 | 1433 | 1433 | 3703 |
| # classes, q | 6 | 3 | 7 | 7 | 6 |

Table 1: The statistics for the citation dataset used in node classification, where each data has the hyperedge either as a co-authorship relation (papers written by the same author) or as a co-citation relation (papers cited by the same paper).

**Additional Baselines.** We additionally discuss the recent methods HypeBoy (Kim et al., 2024) and TriCL (Lee & Shin, 2023), which were developed concurrently or after our work. **Table** 2 presents a performance comparison of these methods. The results indicate that all three methods perform comparably, with each method excelling in different datasets. PhyGCN generally ranks as either the best or second-best method across various datasets and dataset splits. Notably, HypeBoy necessitates extensive softmax computation for nodes against all other nodes, resulting in significant GPU memory consumption and limited scalability to large datasets. Conversely, PhyGCN's self-supervised learning framework is lightweight and scalable, requiring minimal computational resources. TriCL, on the other hand, involves several hyperparameters that need tuning for each dataset. In contrast, PhyGCN only requires tuning the number of layers for different datasets.

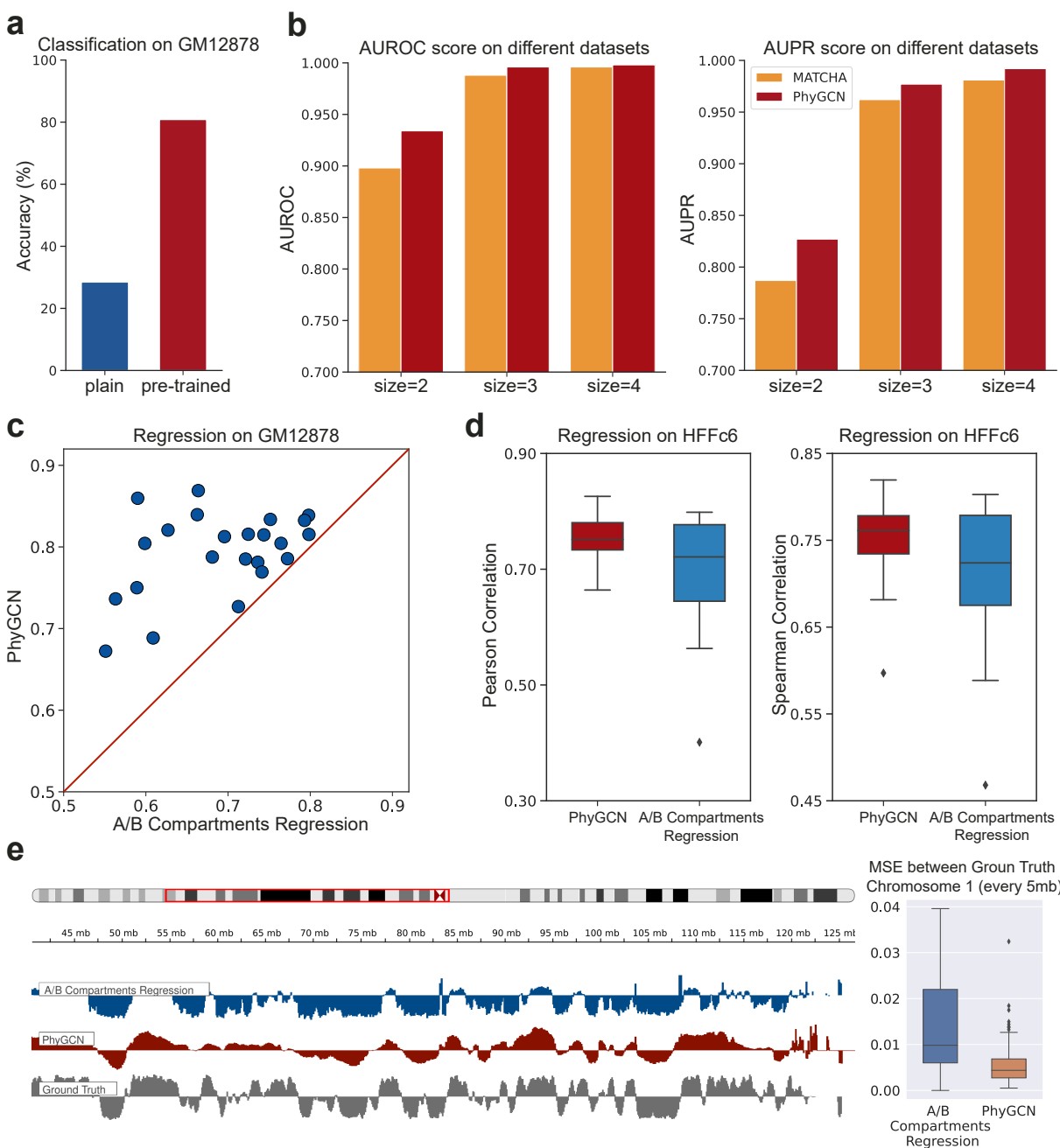

Figure 4: Application to the SPRITE data of chromatin interactions. **a.** Classification accuracy for subcompartment labels of the GM12878 cell line. **b.** AUROC and AUPR scores for predicting multi-way chromatin interactions, where we compare our method with MATCHA (Zhang & Ma, 2020) across hyperedges of sizes 2, 3, and 4. **c.** Regression results on the GM12878 cell line, compared against the baseline A/B Compartments. **d.** Regression results on the HFFc6 cell line, where we compared our transferred PhyGCN results with the baseline A/B Compartments. **e.** Browser shot on chromosome 1 for Repli-seq prediction task (regression on HFFc6). The boxplot shows the MSE distribution between the ground truth and the result from PhyGCN or the baseline. MSE is calculated within each region of 5 Mb.

| Dataset | Model | Training Ratio | | | | | |
|---|---|---|---|---|---|---|---|
| | | 0.5% | 1% | 2% | 4% | 8% | 16% |
| Citeseer | HypeBoy | **48.87 ± 10.3** | **56.88 ± 7.4** | **66.33 ± 3.1** | **68.84 ± 1.5** | 69.89 ± 1.2 | 71.75 ± 1.0 |
| | TriCL | 39.84 ± 9.5 | 48.01 ± 7.7 | 54.31 ± 4.2 | 60.25 ± 2.7 | 64.24 ± 1.1 | 68.91 ± 1.6 |
| | PhyGCN | 47.35 ± 8.4 | 54.22 ± 7.2 | 62.87 ± 4.3 | 67.02 ± 1.0 | **70.07 ± 1.3** | **71.62 ± 1.0** |
| Cora (co-authorship) | HypeBoy | **52.60 ± 6.2** | 60.26 ± 7.6 | 68.96 ± 1.3 | 72.29 ± 1.5 | 73.31 ± 1.3 | 74.00 ± 1.2 |
| | TriCL | 50.94 ± 6.7 | **63.51 ± 6.3** | **70.58 ± 3.6** | **76.71 ± 1.3** | **78.82 ± 0.7** | **80.80 ± 0.8** |
| | PhyGCN | 49.43 ± 6.5 | 60.75 ± 5.8 | 67.21 ± 3.6 | 71.86 ± 1.9 | 75.93 ± 1.3 | 78.39 ± 0.8 |
| Cora (co-citation) | HypeBoy | 47.17 ± 3.9 | 56.77 ± 4.5 | 62.83 ± 5.1 | 67.80 ± 3.1 | 69.65 ± 2.2 | 71.59 ± 2.5 |
| | TriCL | **49.75 ± 7.2** | 56.29 ± 5.9 | **66.27 ± 4.3** | **73.23 ± 1.7** | **77.72 ± 1.3** | **80.42 ± 1.3** |
| | PhyGCN | 48.82 ± 8.2 | **58.46 ± 5.5** | 65.71 ± 4.5 | 72.35 ± 3.2 | 77.33 ± 1.5 | 79.34 ± 0.8 |
| DBLP | HypeBoy | 85.12 ± 0.6 | 85.84 ± 0.2 | 86.07 ± 0.1 | 86.16 ± 0.2 | 86.26 ± 0.3 | 86.27 ± 0.2 |
| | TriCL | **87.99 ± 0.7** | **89.06 ± 0.3** | **89.95 ± 0.2** | **90.37 ± 0.1** | **90.61 ± 0.1** | **90.70 ± 0.1** |
| | PhyGCN | 84.69 ± 0.6 | 86.12 ± 0.4 | 86.96 ± 0.2 | 87.80 ± 0.2 | 88.19 ± 0.2 | 88.23 ± 0.1 |
| PubMed | HypeBoy | 67.84 ± 5.1 | 69.81 ± 3.0 | 69.52 ± 3.4 | 71.02 ± 1.2 | 71.48 ± 0.9 | 71.45 ± 0.8 |
| | TriCL | 69.57 ± 7.8 | 75.10 ± 3.4 | 78.96 ± 1.9 | **80.43 ± 0.8** | 81.68 ± 0.5 | 82.04 ± 0.5 |
| | PhyGCN | **74.66 ± 3.9** | **78.00 ± 1.6** | **79.09 ± 1.6** | 79.96 ± 0.9 | **82.08 ± 0.6** | **82.84 ± 0.5** |

Table 2: Test accuracy for different datasets at different data split ratio (training data % = 0.5% , 1% , 2% , 4%, 8%, 16%). We report mean test accuracy ± standard deviation among 10 random train-test split for each split ratio. We **underscore** the best performing method and underline the second-best performing method.

## 4.2 Exploring multi-way chromatin interaction

To further demonstrate the advantages of PhyGCN, particularly its potential for advancing discoveries in biological data, we applied it to multi-way chromatin interaction datasets. Such datasets reflect simultaneous interactions involving multiple genomic loci within the same nuclei, providing an opportunity to probe higher-order genome organization within a single nucleus. However, due to the limitation of current mapping technology and the exponentially growing combination space of genomic loci, these datasets tend to be extremely sparse and noisy. We evaluated our method on a node classification problem on the SPRITE (Quinodoz et al., 2018) data from the GM12878 cell line, where the aim is to predict the Hi-C subcompartment label for each genomic bin. The details of the experimental setup, including the tasks, baselines, and datasets, can be found in Appendix B.3.

**Why pre-training?** We first evaluated how the pre-training scheme improves the learned representations for the downstream tasks with a node classification problem on the SPRITE data. In this task, we aim to predict the Hi-C subcompartment label for each genomic bin and we split the data by chromosome indices. **Fig. 4a** shows the results of the node classification problem. Without pre-training, our "plain" model is directly trained on the subcompartment labels to make predictions. With self-supervised learning that extracts the hyperedge information, the pre-trained model performs much better on the cross-chromosome subcompartment label prediction task.

**Why the hypergraph convolutional network?** In **Fig. 4b**, we investigate the effectiveness of PhyGCN's hypergraph convolutional architecture in learning multi-way chromatin interactions of variable sizes, which is the pre-training task of the model. Specifically, we compared PhyGCN against MATCHA (Zhang & Ma, 2020), which uses an autoencoder to generate node representations. As in the same setting in Zhang & Ma (2020), we consider hyperedges with the size of 2, 3, and 4 and use the AUROC (area under the receiver operating characteristic) and AUPR (area under precision-recall) scores as the evaluation metrics. We found that, with the hypergraph convolutional architecture, PhyGCN outperforms MATCHA on the three different hyperedge sizes with a small fraction of training data provided. It should be noted that the hyperedge prediction task aims to provide a pre-trained model for downstream tasks. To further evaluate the effectiveness of our base model, we performed several other hyperedge prediction tasks as a thorough ablation study. The results, detailed in Appendix B.5, confirm that the introduced convolutional architecture enables the model to consistently and stably capture hyperedge information across different datasets, providing a better pre-training stage.

**Comparison with previous methods.** To investigate whether our pre-trained model captures informative patterns of multi-way chromatin interactions, we conducted a downstream regression task to predict the DNA replication timing of each genomic bin using corresponding embeddings as input. We fit the signals on even

or odd chromosomes given the training labels on odd or even chromosomes. As a baseline, we used A/B compartment scores (Lieberman-Aiden et al., 2009). The results are shown in **Fig. 4c**, where the $x$-axis represents the Pearson correlation score of A/B compartment scores and Repli-seq signals, and the $y$-axis denotes the Pearson correlation score of the output of PhyGCN and ground truth labels. Each data point on **Fig. 4c** corresponds to a chromosome. Based on the downstream tasks, we found that with pre-training on the hyperedge interactions, our model can effectively capture the underlying patterns of 3D genome organization, generating more informative embeddings that are relevant to important biological functions such as DNA replication.

**Transferring knowledge to a different hypergraph structure.** We further explored the potential of PhyGCN on learning and adapting to new hypergraph structures. Specifically, we evaluated our method on transferring the association between multi-way chromatin interactions and DNA replication timing learned from the SPRITE data of the GM12878 cell line to that of the HFFc6 cell line. In this task, the node set (genomic loci) remain the same across two cell lines, while the observed hyperedges among those nodes (multi-way chromatin interactions) have drastically changed. However, since the label we try to predict is associated with the genome structure that can be depicted by the higher-order chromatin interactions, we reason the transfer learning across hypergraphs would be feasible. To achieve this, we first pre-trained our base model and the attention network to predict multi-way chromatin interactions of the GM12878 cell line using the self-supervised learning task. We then fixed the base model and fine-tune the fully connected layer to predict replication timing of genomic loci in the GM12878 cell line. Next, with the attention network fixed, we fine-tuned the base model with the task of predicting multi-way chromatin interactions in the HFFc6 cell line using the hypergraph constructed from the SPRITE data of HFFc6 following the same SSL setting as the dataset of the GM12878 cell line. With an additional regularizer on the base model weights, we expect our base model to learn the subtle knowledge in hypergraph structural differences and combined with the fixed fully connected layer to make accurate predictions of Repli-seq signals in HFFc6 cell line. The results are shown in **Fig. 4d**, where we use boxplots to show the Pearson and Spearman correlation scores on each chromosome and compare them with the A/B compartment score calculated from the contact of SPRITE data in HFFc6. In **Fig. 4e**, we show an example from chromosome 1 where we compared the ground truth Repli-seq signals to the predictions from A/B compartments regression and PhyGCN with PhyGCN reaching an overall smaller MSE. Together, our results demonstrate that PhyGCN can better learn the representation based on multi-way chromatin interactions and utilize it to make downstream inferences.

**Additional note.** Our problem setting is unique and different from the traditional inductive learning setting. The traditional inductive learning setting involves inference or transfer learning on a new graph with new nodes and new edges/hyperedges among them. In our chromatin interaction network, across different cell lines, the nodes remain the same (as they are the same genomic region and correspond to the same DNA sequence), but the hyperedges among them have changed in different cell lines.

### 4.3 Application to predicting polypharmacy side effect

We also apply PhyGCN on the polypharmacy side effect dataset and demonstrate the advantages of learning such data as a hypergraph rather than a graph. Typically, the association between drugs and side effects is represented as a pairwise drug-side-effect network. However, during disease treatment, multiple drugs are often used in combination, and undesirable combinations can cause side effects not known to either individual drug in the combination (polypharmacy side-effect) (Zitnik et al., 2018). Recent works like Zitnik et al. (2018); Nováček & Mohamed (2020) have used GNNs to predict polypharmacy side-effects with noteworthy performance. These methods treat drugs as nodes and side effects associated with a drug pair as the attributes for the corresponding edge. However, the number of unique side effects is comparable to or even exceeds the number of unique drugs, making it inefficient to model the data as multiple graphs for each side effect. Furthermore, real-world data likely involve drug combinations with more than two drugs that cause side-effects, which GNN methods may not be able to handle. In contrast, hypergraph methods like PhyGCN can easily adapt to such datasets. Since the polypharmacy side-effect data involves interactions beyond pairwise, it can be naturally learned as a hypergraph with drugs and side effects as the nodes. We investigated how PhyGCN learns such interactions and compared it with state-of-the-art methods, including graph-based methods such as Decagon (Zitnik et al., 2018) and ComplEX (Nováček & Mohamed, 2020), and hypergraph-based method Hyper-SAGNN (Zhang et al., 2020). Further details are provided in Appendix B.4.

In Zitnik et al. (2018); Nováček & Mohamed (2020), only side-effects known to be associated with more than 500 drug combinations were used for test data. In addition to this original setting, we report results on random data splits where any side effect can be used for testing. We presented the results in **Fig. 5a**. In terms of AUROC scores, methods based on hypergraph modeling of the data (using Hyper-SAGNN (Zhang et al., 2020) and our PhyGCN) yield better performance than those based on graph modeling (using Decagon (Zitnik et al., 2018) and ComplEX (Nováček & Mohamed, 2020)) in both the original and random settings. Moreover, our algorithm PhyGCN achieves the best performance on all datasets in both settings.

We conducted additional experiments to investigate how PhyGCN benefits from learning the data as a hypergraph by testing it on different types of data splits. Specifically, we considered interactions that involve a known pairwise relationship as "expected" and those that do not involve any known relationship as "unexpected" in the original dataset, as some side-effects are already known to be associated with a single drug (see Appendix B.4). We empirically studied how the baselines and our method performed when testing only on expected or unexpected data. As shown in **Fig. 5a**, GNN methods such as Decagon and ComplEX showed a drop in performance from "expected" to "unexpected", while PhyGCN maintained a consistent performance on both types of data. As the unexpected polypharmacy side-effect triplet does not contain any potential pairwise interactions (while "expected" ones have), the "unexpected" test data serves as a useful indicator of how well a method captured such high-order interactions. The consistency of PhyGCN and Hyper-SAGNN on the different data splits further demonstrates the advantage of modeling complex interactions as a hypergraph. Additionally, our results showed that PhyGCN outperformed Hyper-SAGNN, indicating its superior capability to learn from hypergraphs.

**Enhanced negative sampling strategy.** To investigate the effect of negative sampling on the polypharmacy side effect dataset, we conducted an experiment to compare the traditional random negative sampling method with our proposed negative sampling strategy. The traditional approach (Zitnik et al., 2018; Nováček & Mohamed, 2020) generates negative samples by randomly sampling two drugs (nodes) that do not cause the specific side effect (node) in the training data. These sampled drugs mostly do not cause any side effect, simplifying the prediction so that a model which can predict if a drug pair causes ANY side effect will have satisfactory performance. Therefore, we hypothesize that this strategy may lead to a model that learns the bias of the likelihood that a drug combination causes any side effect rather than the specific side effect in question. To address this, we propose a stronger criterion for negative sampling (see Section 3.3 for details), which involves sampling a drug combination that is known to cause other side effects and combining it with the given side effect as a negative sample. Our enhanced negative sampling strategy, instead of sampling random drug entities, selects an incorrect pair of drugs known to cause other side effects and combines it with the given side effect that it does not cause, therefore increasing the prediction difficulty. This strategy forces the model to differentiate between the positive and negative samples and learn the subtle differences between them. The results of our experiment are presented in **Fig. 5b**. We found that our proposed negative sampling strategy poses a greater challenge for all methods to perform well, with an average test accuracy below 80%. However, we observed that PhyGCN outperforms the baselines and performs stably well across different data splits. This strongly suggests that PhyGCN is better able to grasp the complex interactions without bias, unlike the baselines. We further present detailed examples in **Fig. 5c** to demonstrate the expected and unexpected side effect, and our enhanced negative sampling strategy. Together, the results of this experiment highlight the importance of careful negative sampling in achieving good performance in learning hypergraphs.

**Additional note on the two negative sampling strategy.** We reason our negative sampling strategy resembles a harder task as supported by lower AUROC when evaluating performance. The task for predicting drug pair side effects can actually be seen as a two-step task: 1. A binary classification task, of whether a drug pair can cause any side effect or not; 2. A multi-class classification task, of what specific side effects a drug pair can cause if it will cause side effects. The first task due to the inherent imbalance in the dataset can result in much higher AUROC scores for all methods, and the original negative sampling strategy would be biased towards that. The second task is harder and is what our enhanced negative sampling strategy is trying to highlight.

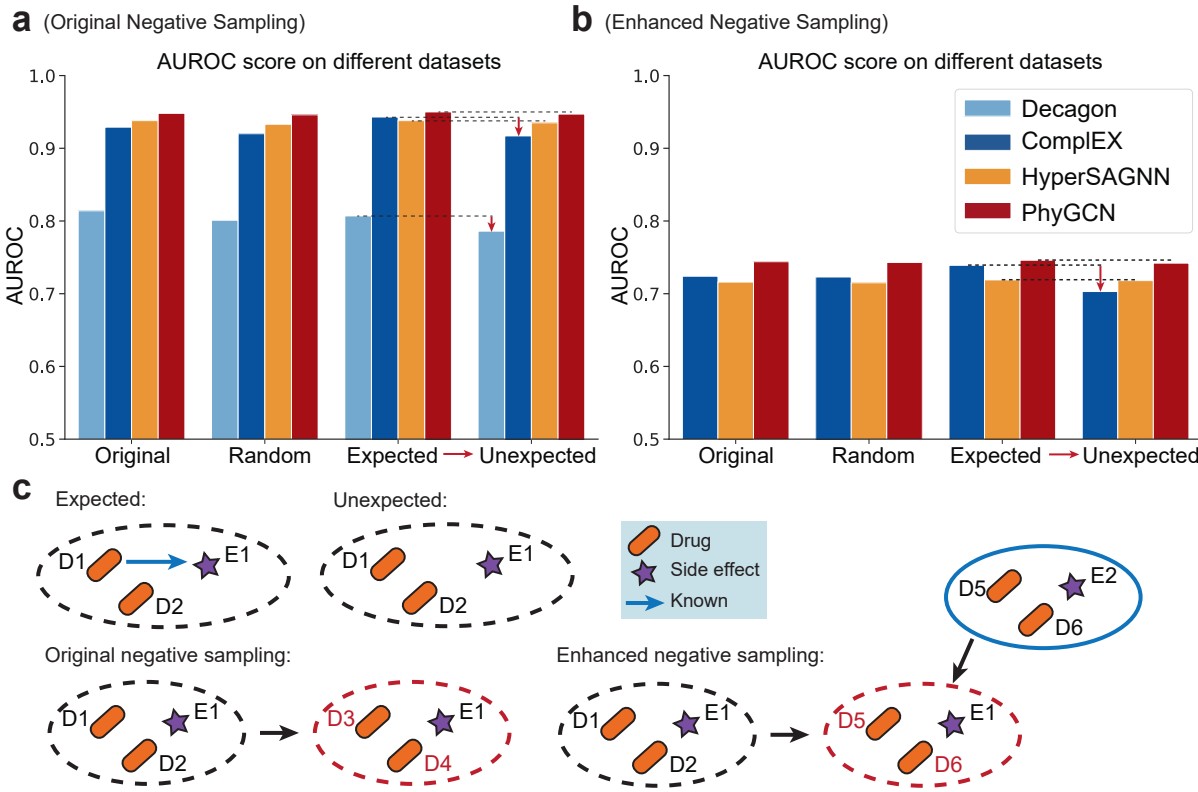

Figure 5: Application to the polypharmacy side effects dataset. **a.** We display the AUROC score for polypharmacy side effect prediction across different data splits, distinguishing between performances on "expected" and "unexpected" polypharmacy side effects. **b.** The AUROC score for polypharmacy side effect prediction across different data splits under an enhanced negative sampling strategy. **c.** We concretely demonstrate our definition of expected/unexpected relations. In the expected interactions, we have prior knowledge that one of the drugs causes the side effect, as indicated by the blue arrow. Furthermore, we illustrate the difference between the original negative sampling and the enhanced negative sampling. The latter selects drug pairs that generate other side effects to be combined with the current example. More discussions on the results in this figure are provided in Section 4.3.

## 5 Discussion

In this paper, we introduced PhyGCN, a hypergraph convolutional network model with a self-supervised pre-training scheme, designed to effectively capture higher-order information of hypergraphs. We demonstrated the effectiveness of PhyGCN on node classification tasks of benchmark citation networks and conduct comprehensive studies on a multi-way chromatin interaction dataset, highlighting the contribution of each component to capturing informative interaction patterns. We also showed the advantages of modeling polypharmacy side effects as hypergraphs and propose an improved negative sampling scheme to evaluate models with less bias. With its ability to model higher-order interactions in various graph-structured data, such as social networks with group interactions, PhyGCN has great potential for a wide range of applications. Many real-world graph-structured data possess higher-order interactions. For example, social networks have group interactions that are inherently hypergraph-structured, and decomposing them into pairwise interactions may lose valuable information. By modeling such graph data as hypergraph, PhyGCN can provide valuable node representations by extracting information from the structure. This offers opportunities for leveraging the method in various applications, from drug discovery to social network analysis, where complex, multi-way interactions play a critical role.

**Acknowledgments**

We would like to thank the anonymous reviewers and the editor for their helpful comments. Y. Deng and Q. Gu were supported in part by the National Science Foundation IIS-1855099 and IIS-1903202. J. Ma was supported, in part, by National Institutes of Health Common Fund 4D Nucleome Program grant UM1HG011593; National Institutes of Health Common Fund Cellular Senescence Network Program grant UH3CA268202; National Institutes of Health grants R01HG007352 and R01HG012303; a Guggenheim Fellowship from the John Simon Guggenheim Memorial Foundation; a Google Research Collabs Award; and a Single-Cell Biology Data Insights award from the Chan Zuckerberg Initiative. P. Xu was supported in part by the National Science Foundation (DMS-2323112) and the Whitehead Scholars Program at the Duke University School of Medicine. R. Zhang was additionally supported by funding from the Eric and Wendy Schmidt Center at the Broad Institute of MIT and Harvard.

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

## A    Additional Related Work

Interactions naturally occur among objects in real-world datasets. The pairwise interactions are conveniently modeled as graphs, facilitating their analysis through deep learning models such as graph neural networks (GNNs) (Li et al., 2016; Kipf & Welling, 2017; Veličković et al., 2018; Xu et al., 2019). GNNs leverage the graph structure to generate representations of unknown nodes for various downstream tasks such as node classification (Yao et al., 2019; Hong et al., 2020), predicting possible unknown interactions between two nodes (Yuan & Bar-Joseph, 2020), and classification of graphs (Zhang et al., 2018). However, real-world data often exhibit relationships that are more complex than pairwise interactions; higher-order interactions involve more than two nodes. For example, in the Cora co-citation network (McCallum et al., 2000), multiple papers are connected by citations from a common paper. Similarly, in the drug side-effect network (Zitnik et al., 2018), interactions may exist among multiple drugs and side effects. Such graph structures with higher-order interactions defines as **hypergraph** (Bretto, 2013), extending the concept of conventional graphs with pairwise interactions. In a hypergraph, these interactions are termed as hyperedges. Being a more complex data structure, hypergraphs contain richer information comapred to graphs with only pairwise edges. However, this complexity poses challenges to classical GNN models, which requires the development of new methods to represent and learn the interactions among arbitrarily many nodes instead of just pairwise.

Reflecting on the well-explored domains GNNs, a mainstream approach for utilizing graph structures is Graph Convolutional Networks (GCNs) (Kipf & Welling, 2017). TGCNs aim to learn node representations by aggregating information from neighboring nodes and combining it with the node's own information. A detailed illustration of GCNs is Section 2. While GCNs have proven effective in representation learning for graphs, the difference in structures between graphs and hypergraphs poses a challenge when adapting GCNs to hypergraphs, especially in capturing the additional information brought by the hyperedges. A straightforward approach to tackle this challenge is to transform the hypergraph into a conventional pairwise graph through clique expansion (Sun et al., 2008), then applying standard GCN methods. Essentially, this entails treating each pair of nodes within a hyperedge as an edge in conventional graphs. Analogous to such expansion, simple convolutional architectures for hypergraphs have been proposed and used in several works (Feng et al., 2019; Bai et al., 2021). Feng et al. (2019) is the first work to introduce a simple neural network structure with a convolutional layer for hyperedges.

Several other works aimed to enhance the performance of hypergraph GCNs by altering the aggregation and combination of neighboring node information. Technically, this variety stems from different designs of the hypergraph's Laplacian matrix (elaborated in Section 2). Yadati et al. (2019) also approximated hyperedges with a set of pairwise edges, proposing the HyperGCN scheme that achieves the best result and FastHyperGCN that trains faster albeit with compromised performance. Subsequently, to better capture hyperedge information, Dong et al. (2020) proposed Hypergraph Networks with Hyperedge Neurons (HNHN), a hypergraph convolutional network applying nonlinear activation functions to both nodes and hyperedges. This model computes embeddings for nodes and hyperedges. However, under varied train-test data splits on multiple hypergraph data, these methods could exhibit instability and perform worse than a common GCN learning on hypergraphs expanded as regular graphs. Other attempts to adapt graph architectures to hypergraphs either achieved similar or slightly improved performance compared to HyperGCN (Arya et al., 2020; Bai et al., 2021; Yi & Park, 2020) or were evaluated under different conditions such as a larger amount of training data (Sun et al., 2021).

Expansions like clique expansion assume a hyperedge's decomposability, implying that any subset of nodes in a hyperedge can form another hyperedge. However, this assumption becomes unreasonable when a hyperedge contains nodes of different types. For instance, a hyperedge may represent a triplet relationship between a user, drug, and reaction. Treating such a hyperedge as pairwise edges could be detrimental to downstream tasks. This issue was similarly noted in Tu et al. (2018), where the authors attempted to model the hypergraph with a simple encoder-decoder model. Yet, as evidenced in our empirical analysis for the multiway chromatin interaction data, such a model does not match the capability of a deep hypergraphgraph convolutional model.

While relatively few works explore self-supervised training or pre-training on hypergraphs, the field of graphs has seen many mature works. Comprehensive studies (Hu et al., 2020a; You et al., 2020b; Wu et al., 2021; Jin et al., 2022) have introduced and examined multiple self-supervised tasks for graphs at both the node and graph level. Additionally, some works (Hu et al., 2020b; Hwang et al., 2020; Hao et al., 2021; Sun et al., 2020) were tailored for specific settings, employing one or more similar self-supervised tasks to enhance model performance and adapt to chosen scenarios. In another subfield of self-supervised training, contrastive learning for GNNs has also been explored in recent studies (Hassani & Khasahmadi, 2020; You et al., 2020a; Qiu et al., 2020).

## B  Experimental Details

### B.1  Model Parameter Setting in Pre-training

For PhyGCN, we consider three hypergraph convolutional layers, each with a uniform hidden layer size of 128 and an output embedding size of 64. In the multi-head attention layer used for pre-training, we set the number of heads to 8, aligning with setting in Zhang et al. (2020). The number of training epochs is set to be 300, and the training batch of the pre-training task contains 32 positive samples while we generate 5 negative samples with regard to each positive sample. We use Adam optimizer of learning rate 0.001 with weight decay 5e-4. The dropout rate is set to 0.5, and the coefficient for dropedge is set to 0.3: masking 30% of the values of the adjacency matrix $\widehat{\mathbf{A}}$ at each training iteration.

### B.2  Supplementary Information for the Citation Networks Data

**Baselines.**   We compare our approach with two recent baselines on hypergraph neural networks alongside an additional baseline derived from modifying the graph convolutional network:

- HyperGCN, proposed by Yadati et al. (2019), represents their best performing model. This method incorporates node features when calculating the Laplacian to improve performances. However, such computation significantly increases the computation cost for the method.

- HNHN, introduced by Dong et al. (2020), divides the aggregation process into two steps, computing embeddings for hyperedges and nodes separately.

- GCN, here we expand the hypergraph into graphs with pairwise edges through clique expansion and apply a straightforward GCN model Kipf & Welling (2017) for prediction.

**Training.**   All baselines are trained using the Adam optimizer Kingma & Ba (2014) as specified in their respective papers. The selected hyperparameters for each baseline also adhere to those suggested in the respective papers. For PhyGCN, we use SGD of learning rate 0.003 with momentum 0.9 for the node classification task.

**Detailed Empirical Results**   Detailed results of the average test accuracy $\pm$ standard deviation for all models on all datasets are shown in **Table** 3. We found that PhyGCN outperforms all baselines on each dataset regardless of the training data ratios. On smaller hypergraphs like Cora and Citeseer, PhyGCN is able to leverage more information with self-supervised learning and outperforms all baselines by a large margin. Similarly, in scenarios with low training data ratio on large networks, PhyGCN maintains remarkable robustness and exemplary performance, while other hypergraph methods such as HyperGCN and HNHN falter. An ablation study of the pre-training scheme presents results of our hypergraph convolutional network both with and without pre-training. A direct comparison shows the substantial benefit of pre-training. For instance, on the Cora dataset, where the plain model underperforms when compared to the vanilla GCN, pre-training enables the hypergraph convolutional network to better capture higher-order information, and significantly outperforming GCN.

**Ablation Study on Base Network.**   In table 4, we additionally investigate the effect of changing our base hypergraph convolutional network to one that is used in HyperSAGNN (Zhang et al., 2020) (encoder-based). We note that, HyperSAGNN is a method designed for hyperedge prediction, and is therefore only considered as a baseline on hyperedge prediction tasks. The investigation on downstream tasks such as

| Dataset | Model | Training Ratio | | | | | |
|---|---|---|---|---|---|---|---|
| | | 0.5% | 1% | 2% | 4% | 8% | 16% |
| Citeseer | HyperGCN | 30.71 ± 7.0 | 41.02 ± 10.6 | 51.50 ± 8.5 | 63.10 ± 4.6 | 68.93 ± 2.1 | 71.10 ± 1.2 |
| | HNHN | 26.33 ± 7.5 | 28.61 ± 5.9 | 34.65 ± 6.9 | 45.35 ± 6.5 | 53.62 ± 5.2 | 58.12 ± 2.1 |
| | GCN | 32.66 ± 10.4 | 44.02 ± 9.7 | 53.91 ± 6.4 | 62.54 ± 3.7 | 69.70 ± 1.3 | 71.47 ± 1.1 |
| | PhyGCN (plain) | 32.74 ± 6.2 | 38.99 ± 6.5 | 49.21 ± 0.5 | 56.72 ± 3.2 | 59.94 ± 3.1 | 65.58 ± 1.3 |
| | PhyGCN | **47.35 ± 8.4** | **54.22 ± 7.2** | **62.87 ± 4.3** | **67.02 ± 1.0** | **70.07 ± 1.3** | **71.62 ± 1.0** |
| Cora (co-authorship) | HyperGCN | 33.08 ± 8.6 | 39.50 ± 4.7 | 49.24 ± 4.8 | 57.36 ± 5.2 | 65.38 ± 3.8 | 70.39 ± 3.5 |
| | HNHN | 30.48 ± 5.5 | 38.23 ± 10.4 | 53.53 ± 4.0 | 63.51 ± 3.1 | 70.01 ± 2.2 | 74.51 ± 1.8 |
| | GCN | 38.92 ± 9.3 | 54.34 ± 7.4 | 60.35 ± 5.3 | 69.00 ± 2.2 | 73.48 ± 1.3 | 76.01 ± 0.8 |
| | PhyGCN (plain) | 38.21 ± 7.6 | 50.21 ± 5.6 | 59.66 ± 5.7 | 68.53 ± 2.6 | 71.73 ± 1.4 | 73.99 ± 1.5 |
| | PhyGCN | **49.43 ± 6.5** | **60.75 ± 5.8** | **67.21 ± 3.6** | **71.86 ± 1.9** | **75.93 ± 1.3** | **78.39 ± 0.8** |
| Cora (co-citation) | HyperGCN | 33.06 ± 6.2 | 37.31 ± 7.3 | 45.97 ± 6.9 | 56.72 ± 5.6 | 63.67 ± 3.7 | 65.12 ± 4.0 |
| | HNHN | 26.16 ± 4.5 | 28.44 ± 4.7 | 38.51 ± 7.5 | 55.60 ± 3.3 | 64.13 ± 1.8 | 65.29 ± 1.6 |
| | GCN | 39.52 ± 9.7 | 48.37 ± 6.8 | 58.72 ± 7.0 | 71.15 ± 3.5 | 74.28 ± 1.2 | 76.83 ± 1.3 |
| | PhyGCN (plain) | 33.37 ± 6.6 | 38.31 ± 5.6 | 48.77 ± 5.5 | 62.28 ± 3.2 | 68.51 ± 1.3 | 76.18 ± 1.0 |
| | PhyGCN | **48.82 ± 8.2** | **58.46 ± 5.5** | **65.71 ± 4.5** | **72.35 ± 3.2** | **77.33 ± 1.5** | **79.34 ± 0.8** |
| DBLP | HyperGCN | 65.35 ± 4.3 | 66.30 ± 6.2 | 63.79 ± 4.7 | 65.84 ± 6.4 | 70.72 ± 5.2 | 73.93 ± 4.5 |
| | HNHN | 74.41 ± 1.9 | 79.86 ± 0.9 | 82.19 ± 0.4 | 83.73 ± 0.2 | 84.66 ± 0.3 | 85.05 ± 0.2 |
| | GCN | 81.11 ± 2.9 | 83.65 ± 2.0 | 85.32 ± 0.7 | 85.84 ± 0.5 | 86.54 ± 0.2 | 86.82 ± 0.3 |
| | PhyGCN (plain) | 81.90 ± 0.6 | 83.71 ± 0.4 | 85.24 ± 0.2 | 86.52 ± 0.2 | 86.95 ± 0.3 | 87.43 ± 0.2 |
| | PhyGCN | **84.69 ± 0.6** | **86.12 ± 0.4** | **86.96 ± 0.2** | **87.80 ± 0.2** | **88.19 ± 0.2** | **88.23 ± 0.1** |
| PubMed | HyperGCN | 67.84 ± 6.3 | 72.29 ± 5.4 | 77.10 ± 2.1 | 79.63 ± 1.0 | 82.05 ± 1.0 | **83.56 ± 0.6** |
| | HNHN | 63.61 ± 2.9 | 68.86 ± 3.9 | 72.24 ± 3.2 | 76.49 ± 1.6 | 79.72 ± 1.2 | 80.74 ± 0.9 |
| | GCN | 70.29 ± 4.4 | 74.31 ± 2.8 | 77.92 ± 1.5 | 79.66 ± 1.0 | 80.81 ± 0.6 | 81.51 ± 0.6 |
| | PhyGCN (plain) | 70.96 ± 3.7 | 75.07 ± 2.2 | 78.01 ± 1.0 | 79.84 ± 0.5 | 81.13 ± 0.9 | 82.42 ± 0.4 |
| | PhyGCN | **74.66 ± 3.9** | **78.00 ± 1.6** | **79.09 ± 1.6** | **79.96 ± 0.9** | **82.08 ± 0.6** | 82.84 ± 0.5 |

Table 3: Test accuracy for different datasets at different data split ratio (training data % = 0.5% , 1% , 2% , 4%, 8%, 16%). We report mean test accuracy ± standard deviation among 10 random train-test split for each split ratio.

node classification is considered as an ablations study on the benefits of different base networks for feature learning. In the later sections, we further include a comprehensive discussion on the effect of the two different base networks (encoder vs. hypergraph convolution) on hyperedge prediction (**Table** 10 and 11).

| Base | Training Ratio | | | | | |
|---|---|---|---|---|---|---|
| | 0.5% | 1% | 2% | 4% | 8% | 16% |
| Encoder | 45.89 ± 7.9 | 51.76 ± 7.3 | 61.04 ± 4.1 | 66.85 ± 1.9 | 69.97 ± 1.3 | 71.55 ± 0.8 |
| Conv | **47.35 ± 8.4** | **54.22 ± 7.2** | **62.87 ± 4.3** | **67.02 ± 1.0** | **70.07 ± 1.3** | **71.62 ± 1.0** |

Table 4: Test accuracy for Citeseer at different data split ratio (training data % = 0.5% , 1% , 2% , 4%, 8%, 16%). We report mean test accuracy ± standard deviation among 10 random train-test split for each split ratio.

### B.3 Supplementary Information for the Multiway Chromatin Interaction Data

**Baselines.** For the hyperedge prediction task on multi-way chromatin interaction, we evaluated against a recent work, MATCHA (Zhang & Ma, 2020), which employs hypergraph representation learning techniques on a specially constructed hypergraph from multi-way chromatin interaction data, aimed at denoising the data and making *de-novo* predictions of yet undetected multi-way chromatin interactions. For the baseline of **Fig. 4c** and **Fig. 4d**, we used A/B compartment scores (Lieberman-Aiden et al., 2009), a 3D genome features that is known to correlate with DNA replication timing. The A/B compartment scores are calculated using the Cooler software (Abdennur & Mirny, 2020) by taking the SPRITE contact map as input.

**Dataset.** To systematically evaluate if our proposed new approach could improve the analysis of multi-way chromatin interaction data, we use the SPRITE data (Quinodoz et al., 2018) from the GM12878 lymphoblastoid human cell line following the same processing procedure in MATCHA (Zhang & Ma, 2020). Specifically, genomic bins at the 1Mb resolution are considered as nodes, and interactions involving multiple loci are treated as hyperedges. In **Fig. 4a**, a split by chromosome indices is performed, where nodes odd chromosome indices constitute the training data, and those from even indices form the test data. For cross-validation, a swapping of the training and test set is conducted, followed by the reporting of the average accuracy for each method. **Fig. 4b** displays a comparison against MATCHA (Zhang & Ma, 2020) under a more challenging training split, constructing the hypergraph from SPRITE data with 20% designated as training set and the

|  | Accuracy |
|---|---|
| Plain | 28.42 % |
| Pre-trained | **80.79 %** |

Table 5: Classification on GM12878.

|  | MATCHA | Our Method |
|---|---|---|
| Size= 2 | (0.898, 0.787) | **(0.934, 0.827)** |
| Size= 3 | (0.988, 0.962) | **(0.996, 0.977)** |
| Size= 4 | (0.996, 0.981) | **(0.998, 0.992)** |

Table 6: Hyperedge prediction on SPRITE data. Size denotes the length of the hyperedge. We report the (AUROC, AUPR) scores.

remaining 80% as test data. **Fig. 4c** and **Fig. 4d**, the replication timing for each genomic bin is quantified using Repli-seq (Marchal et al., 2018).

**Detailed Empirical Results.** **Table** 5 shows the detailed results for **Fig. 4a**. **Table** 6 shows the detailed results for **Fig. 4b**.

### B.4 Supplementary Information for Polypharmacy Side Effect Data

**Baselines.** We compared our approach with several methods on polypharmacy side effect prediction:

- Decagon, proposed by Zitnik et al. (2018) as the initial work that studies the polypharmacy side-effect data.

- ComplEX, proposed in Nováček & Mohamed (2020) that has the best reported performances.

- HyperSAGNN, with the encoder-decoder model for representation learning as proposed in Zhang et al. (2020).

**Dataset.** We use the polypharmacy side-effect dataset (Zitnik et al., 2018), which contains 645 unique drugs and 63,473 drug-drug interactions. Within the hypergraph framework, there are 4,651,131 hyperedges representing the drug-drug side-effect associations, featuring 1,317 unique side effects. The evaluation on side-effect prediction was conducted across four distinct train-test split scenarios:

- *Original*: The same as in Zitnik et al. (2018), where the test set only contains side effects associated with more than 500 drug combinations, whilst side-effects liked with less than 500 drug combinations are allocated to the training set.

- *Random*: All side-effects are utilized for both training and testing, distributed randomly across training and test sets.

- *Expected*: A subset of 30 side-effects, already associated with a single drug, were deemed as expected, leading to 20,787 expected drug-drug side-effect associations. Despite adhering to a random split for training and test data, the "expected" scenario only tests on the expected triplets within the test set.

- *Unexpected*: Similarly, whilst maintaining a random split for training and test data, testing is conducted solely on the unexpected triplets in the test set.

For the side effect categorization task, we consider side effects with known categories from Zitnik et al. (2018), yielding 561 side effects with 37 categories.

**Detailed Empirical Results.** In **Table** 8 and **Table** 9, we show detailed results of the AUROC and AUPR scores for all models on the four different evaluation sets. The AUROC scores in **Table** 8 correspond to **Fig. 4e** and the AUROC scores in **Table** 9 correspond to **Fig. 4f**. The AUPR scores of **Table** 8 showed a similar pattern as AUROC scores, where hypergraph methods tend to maintain their performance on expected and unexpected test data while graph methods exhibit a decline in performance. Similarly, for the

| Model and training scheme | Accuracy |
|---|---|
| Plain | 21.49 % |
| With pre-training, random negative sampling | 33.65 % |
| With pre-training, modified negative sampling | **35.23 %** |

Table 7: Classification of side-effects.

| | Decagon | ComplEX | Hyper-SAGNN | Our Method |
|---|---|---|---|---|
| Original | (0.814, 0.737) | (0.929, 0.914) | (0.938, 0.921) | **(0.945, 0.931)** |
| Random | (0.801, 0.729) | (0.920, 0.903) | (0.933, 0.912) | **(0.943, 0.926)** |
| Expected | (0.807, 0.733) | (0.943, 0.931) | (0.938, 0.924) | **(0.947, 0.932)** |
| Unexpected | (0.786, 0.713) | (0.917, 0.890) | (0.935, 0.918) | **(0.944, 0.922)** |

Table 8: Performances on different data split. Random negative sampling. We report the (AUROC, AUPR) scores for each model on each data split.

AUPR scores of **Table** 9, the enhanced negative sampling strategy brings more challenge to the task and forces the networks to learn the more subtle differences between positive and negative samples.

Our results demonstrated the adeptness of our model in learning higher-order interactions effectively. We also studied how learning within the hypergraph structure could improve our model's performance on tasks pertinent to the nodes (drugs, side effects). We used the side effect category information from Zitnik et al. (2018), where a proportion of the side effects within the network were classified into high-level categories. As shown in **Table** 7, with pre-training on the hypergraph structure, our model exhibited enhanced categorization capabilities, even in the absence of additional information. Furthermore, the performance metric could be improved through the implementation of a better negative sampling strategy.

### B.5 The Base Convolutional Model Learns Hyperedge Information Well

We further evaluated how well the hypergraph convolutional network in PhyGCN captures the hyperedge information. We conducted evaluation on the the four datasets used in the most current works on hyperedge prediction (Tu et al., 2018; Zhang et al., 2020). These datasets, shown below, have uniform-length hyperedges and no node features or node labels.

- GPS (Zheng et al., 2010): hyperedges represent (user, location, activity) relations.

- MovieLens (Harper & Konstan, 2015): hyperedges represent (user, movie, tag) relations.

- drug: hyperedges represent (user, drug, reaction) relations.

- wordnet (Bordes et al., 2013): hyperedges represent (head entity, relation, tail entity) relations within words.

Hyperedge prediction methods need to use structural information as the features to make predictions. Moreover, these datasets are heterogeneous, meaning that the node types within a hyperedge are different and therefore one could not simply expand the hyperedge into pairwise edges. Details of the experiment results are reported in **Table** 10. We evaluated our base convolutional model against the encoder-decoder model used in Tu et al. (2018); Zhang et al. (2020) for representation learning, using the same attention network from Zhang et al. (2020) for combining the node embeddings and making the prediction. The performance is evaluated by the AUROC and the AUPR score. We found that, with the introduction of convolutional architecture, our model performs either better than or on par with the encoder-decoder model. We note that the performances for the four datasets are already very high with the encoder-decoder model, our base model still exhibit enhanced capability in capturing the hyperedge information.

| | ComplEX | Hyper-SAGNN | Our Method |
|---|---|---|---|
| Original | (0.724, 0.702) | (0.716, 0.702) | **(0.744, 0.731)** |
| Random | (0.723, 0.699) | (0.715, 0.703) | **(0.743, 0.730)** |
| Expected | (0.739, 0.722) | (0.719, 0.709) | **(0.746, 0.732)** |
| Unexpected | (0.703, 0.686) | (0.718, 0.703) | **(0.742, 0.729)** |

Table 9: Performances on different data split. Modified negative sampling. We report the (AUROC, AUPR) scores for each model on each data split.

|  | GPS | MovieLens | drug | wordnet |
|---|---|---|---|---|
| Encoder | (0.930, 0.744) | (0.926, 0.793) | (0.961, 0.888) | (0.890, 0.694) |
| Conv | **(0.941, 0.738)** | **(0.938, 0.801)** | (0.962, 0.890) | **(0.896, 0.710)** |

Table 10: The experiment results for the hyperedge prediction task over the four heterogeneous hypergraph data with uniform-length hyperedges. We report (AUROC, AUPR) to evaluate how well our model learns the structural knowledge. Our convolutional model is compared with the encoder-decoder model used in Zhang et al. (2020), both using the attention layer for prediction.

|  | Citeseer | Cora (co-authorship) | Cora (co-cocitation) | DBLP | PubMed |
|---|---|---|---|---|---|
| Encoder-1 | (0.804, 0.573) | (0.827, 0.626) | (0.809, 0.591) | (0.825, 0.655) | (0.906, 0.722) |
| Encoder-2 | (0.833, 0.612) | (0.798, 0.553) | (0.849, 0.590) | (0.920, 0.727) | (0.919, 0.730) |
| Encoder-3 | (0.803, 0.583) | (0.826, 0.660) | (0.839, 0.607) | **(0.930, 0.773)** | **(0.920, 0.752)** |
| Conv | **(0.866, 0.605)** | **(0.861, 0.657)** | **(0.861, 0.597)** | (0.923, 0.755) | **(0.926, 0.727)** |

Table 11: The experiment results for hyperedge prediction over the five homogeneous datasets with variable-length hyperedges. We report (AUROC, AUPR) similarly. We compare our convolutional model with the encoder-decoder model that takes in three different features. Encoder-1: original features computed as in Zhang et al. (2020). Encoder-2: node features provided by the data. Encoder-3: concatenation of the two.

In **Table** 11, we extended our evaluation to five datasets for the main task of node classification. These datasets are citation networks where each node represents a publication and each hyperedge represents either co-citation or co-authorship relation. These data are different from the above setting, where the hypergraph is homogeneous with variable hyperedge size. We note that, for the previous hyperedge prediction data, the methods typically use features constructed from the hypergraph structure: a detailed formulation of the construction of input features are presented in Sections 2 and 3. However, as these five datasets have node features themselves, we additionally add two modified baselines of the encoder-decoder model for a more comprehensive evaluation. Specifically, we compare with encoder-decoder that takes in the constructed features, the node features, and concatenation of the two. Our model soley utilizes the node feature for these five datasets. We found that our convolutional base model outperforms the three baselines on AUROC score consistently across the five datasets. Such results underscore the potential of our base model to effectively and efficiently utilize both the node features and the structural information of the hypergraph. For the hyperedge prediction task only, the base convolutional model has a stably better performance compared to the simple model used in previous works. Such evaluation confirms that the model learns the structural knowledge well in this pre-training task.

