# OpenReview forum: "Pre-trained Hypergraph Convolutional Neural Networks with Self-supervised Learning"
_TMLR — Accepted by TMLR_

### Review · Reviewer_ySwi · 2024-05-04

**Summary Of Contributions:**

This manuscript proposes a graph neural network pretraining method for hypergraphs. In a hypergraph, each hyperedge involves more than two nodes. The proposed hypergraph neural network represents a hypergraph using an incidence matrix and uses a GCN-style aggregation scheme to encode the graph. To train the hypergraph neural network in a self supervised learning manner, the manuscript proposes hyperedge prediction as the pretraining task, resulting in enhanced node representations. Given a downstream task,  such as node classification, the pretrained node representations are fed into a MLP finetuned for prediction. Experiments on the node classification task show that pretrained node representations can improve the performance of the task.

**Audience:**

Yes

**Broader Impact Concerns:**

NA.

**Claims And Evidence:**

Yes

**Requested Changes:**

Below is my suggestions to improve the manuscript:

(1) Add more GNN or pretraining baselines in the experiments to make the work more solid (if possible).

(2) Add more details on the used datasets.

(3) Add more explanations about the experiment "Transferring knowledge to a different hypergraph structure".

(4) Highlight the difference between the proposed method and existing work (in terms of techniques).

**Strengths And Weaknesses:**

Strengthes:

S1. This manuscript systematically shows how to pretrain hypergraphs and how to use pretrain hypergraph embeddings to enhance the performance of downstream tasks.

S2. Overall, this manuscript is well-structured and well presented, providing a detailed description of the proposed method.

Weaknesses:

W1. The proposed method has limited technical contributions. From a broader view of graph neural network research, the used techniques, such as the link prediction style pretraining, MLP finetuning and negative sampling strategy, come from existing experience.

W2. I have several concerns about the experiments. It seems (from the related work section) that there are some hypergraph pretraining methods in the literature, such as Du et al. (2021) , but they were not selected as baselines. Besides, it is not clear to me how to extend the proposed method to unseen graphs. It seems that in the experiment "Transferring knowledge to a different hypergraph structure", the base model is finetuned using the target data, which is different from the conventional setting of inductive learning for graphs. More explanations or details are necessary.

W3. From the related work section, it seems that there are hypergraph neural networks and hypergraph neural network pretraining methods in the literature. What is the key contribution of the proposed method compared to the existing work? It would be better to highlight the difference between the proposed method and other work.

W4. I also have a concern regarding the used datasets. In my view, the used the citation data (such as DBLP) is a multi-relational graph, where each edge links to nodes and has a label denoting the relation type. Is a multi-relational graph also a hypergraph? It would be be better to add more introductions about the datasets, including what is the data structure and node feature.

W5. The proposed hypergraph neural network may face scalability challenges when dealing with large hypergraphs. It represets the graph using an incidence matrix $H \in R^{N×M}$, with N as the number of nodes and M as the number of hyperedges. In general, the number of edges is much larger than the number of nodes in a graph. The computational cost increases as the number of nodes and hyperedges grows.

---

> ### Author Response · Authors · 2024-06-13
> **Response to the reviewer's comments**
>
> Thank you very much for your helpful comments and suggestions. We answer your questions as follows. We have also revised our paper accordingly to incorporate your suggested changes.
>
> ---
> **Q:** Add more GNN or pretraining baselines in the experiments to make the work more solid (if possible).
>
> **A:** We thank the reviewer for this suggestion. We have included two more recent works related to hypergraph representation learning or pre-training for comparisons, including Hypeboy [2] and TriCL [1]. The additional results are demonstrated in Table 3 of Appendix B.2. Our results indicate that all three methods perform comparably, with each method excelling on different datasets. PhyGCN generally ranks as either the best or second-best method across various datasets and dataset splits. Notably, HypeBoy necessitates extensive softmax computation for nodes against all other nodes, resulting in significant GPU memory consumption and limited scalability to large datasets. Conversely, PhyGCN’s self-supervised learning framework is lightweight and scalable, requiring minimal computational resources. TriCL, on the other hand, involves several hyperparameters that need tuning for each dataset. In contrast, PhyGCN only requires tuning the number of layers for different datasets.
>
> References:
> [1] I’m me, we’re us, and i’m us: Tri-directional contrastive learning on hypergraphs. Lee and Shin. AAAI 2023.
> [2] HypeBoy: Generative Self-Supervised Representation Learning on Hypergraphs. Kim et al. ICLR 2024.
>
> ---
> **Q:** Add more details on the used datasets.
>
> **A:** Thank you for the suggestion. Due to the space limit, we added the details of the datasets in the appendix. In particular, the description of the citation datasets is displayed in Table 1.
>
> Regarding the reviewer’s question on whether citation networks can be considered as hypergraphs, we provide the clarification as follows. In current hypergraph research, citation networks are modeled as hypergraphs where hyperedges contain publications sharing the same citation (co-citation) or the same author (co-authorship). Since multiple publications can cite the same paper or share the same author, they form hyperedges with arbitrary sizes (i.e., variable-sized hyperedge). We have added this clarification in our revision on section 4.1 page 7.
>
> ---
> **Q:** Add more explanations about the experiment "Transferring knowledge to a different hypergraph structure".
>
> **A:** We apologize for the confusion. The reviewer is correct that our problem setting is unique and different from the traditional inductive learning setting. The traditional inductive learning setting involves inference or transfer learning on a new graph with new nodes and new edges/hyperedges among them. In our chromatin interaction network, across different cell lines, the nodes remain the same (as they are the same genomic region and correspond to the same DNA sequence), but the hyperedges among them have changed in different cell lines.
>
> ---
> **Q:** Highlight the difference between the proposed method and existing work (in terms of techniques).
>
> **A:** We have added the discussion of some recent work pointed out by the reviewers and their follow-up work as well in the revised introduction. In particular, [1] studies temporal hypergraphs and uses random walk to automatically extracts temporal, higher-order motifs. [2] studies how the concept of homophily can be characterized in hypergraphs, and proposes a novel definition of homophily to effectively describe HNN model performances. Both works are related to our study on hypergraphs but orthogonal to our focus on the performance improvement by self-supervised learning on hypergraphs. [3] studies self-supervised learning on hypergraphs and utilizes a tri-directional contrastive loss function consisting of node-, hyperedge-, and membership-level contrast. Very recently, [4] studies generative self-supervised learning on hypergraphs and proposed a new task called hyperedge filling for hypergraph representation learning. Note that [2] and [4] are released after our work appeared online. Since [3] and [4] are most related to ours, we conducted additional experiments to compare them without method. Please refer to our answer to your first question to see the details of the experiments.
>
> References:
> [1] CAT-Walk: Inductive Hypergraph Learning via Set Walks. Behrouz et al. NeurIPS 2023.
> [2] Hypergraph neural networks through the lens of message passing: a common perspective to homophily and architecture design. Telyatnikov et al. 2023.
> [3] I’m me, we’re us, and i’m us: Tri-directional contrastive learning on hypergraphs. Lee and Shin. AAAI
> [4] HypeBoy: Generative Self-Supervised Representation Learning on Hypergraphs. Kim et al. ICLR 2024.

---

> > ### Comment · Reviewer_ySwi · 2024-06-26
> > **Thanks for your response**
> >
> > Dear authors:
> >
> > Thanks for your response. The revised version is much better. I have no additional questions.

---

### Review · Reviewer_d2MZ · 2024-05-14

**Summary Of Contributions:**

In this work, the orignal hypergraph-able GNN called Hyper-SAGNN (Zhang 2020) and the corresponding hyperedge prediction task (including the negative sampling strategy for performing SSL) are used as pre-training for improving prediction accuracy on various downstream tasks.
It is shown that the pre-training improves the accuracy.

Furthermore, a few points seem to be original:
- The way to fight vanishing gradient is somehow original but has most liekly been used before elsewhere (it's not an auxiliary loss but close to it, as each layer's representation is concatenated for final decoding).
- Also, for the pretraining, Dropout is replaced with DropHyperEdge (slightly different from the previously known DropEdge).
- A new negative sampling strategy, specific to the hyperedge geometry, is introduced, and studied empirically.

**Audience:**

Yes

**Broader Impact Concerns:**

None.

**Claims And Evidence:**

Yes

**Requested Changes:**

Important points:
- In fig 3, why is SAGNN (Zhang 2020) not presented ? It seems to me it must be, for fair comparison.
- In the experiment section, the size of the pre-training dataset is not mentionned. For instance, when training with 0.5% of the available data, how is the pre-training done ? With the same amount of raw hypergraphs, but using numerous randomized samplings to self-supervise in many ways ? Or, was a larger corpus of hypergraphs used ?
This is a very important point, that must be mentionned explicitly, regardless of the choice you made.
- Why limit to 4% of the data at maximum ?
I guess PhyGCN becomes less good than other models when data is abundant. That's ok. I think it would be more honest, and interesting, to report also larger values of N_train.
- The pre-training loss is not written down in the main text. It should be, with a clear role of the positive/negative samples. This is important because in a first read, it seems the only task is to recover missing Hyperedges (as a pretext task).
- The outcome of the experiment presented in fig 5 is insufficiently commented on. As I read it, it goes like this:
- AUROC means larger is better.
- you perform the pretext task of hyperedge prediction using 4 different methods, for 2 kinds of negative sampling.
- Results are WORSE using your enhanced sampling strategy.
- conclusion: the enhanced sampling is more challenging to figure out, thus, a better way to pre-train.
- side remark: PhyGCN is not affected by the affected/unaffected split, whereas other methods are, a bit, which means that PhyGCN is especially good when hyperedges are not simply two-node edges.
If I'm correct, good for me, but, it'd be better to be slightly more explicit about the reasoning. You never write explicitly what is the task performed by these nets, in ML terms. (I can't read the biology jargon). Also the conclusion you draw is not clear.
This is a pity because as I gather, this original negative sampling strategy is one of the genuinely new ideas presented in the paper.

Moderately important points:
- First read: it's not immediately clear what you mean by variable-sized hyperedge prediction (I mean, why is it non trivial, why and how is attention used/helpful for that?)
Maybe a small comment to outline the intuition would be helpful.
(After looking at the equations again, and remembering that attention can be input-size-independent, it makes perfect sense - but one has to do the reasoning by themselves).
- Negative sampling strategy:
- I do not fully understand the last Negative sampling strategy, (text just a bit before Eq 8). It's not very important, but probably it could be explained more clearly.
- Again, Enhanced negative sampling: figure 5:
honestly, this is still not very clear to me.
- BUT, I did understand when reading "To address this, we propose a stronger criterion for negative sampling, which involves
sampling a drug combination that is known to cause other side effects and combining it with the given
side effect as a negative sample."


Minor points:

- AUROC and AUPR are acronyms and are not defined.
Area Under the ROC curve ?
AUPR : ??

- fine-tuning: it is sometimes slightly unclear whether fine-tuning means training the decoder (h) weights only, or also those of the base model f (and/or those of the attention network g).
It would be better to explicitly state "keeping the base model fixed" or "further training the base model" each time.
Independently from that, I was unable to understand how the results fo fig 4d, 4e were obtained, from the following sentence (which clearly aims at being explicit, but.. I failed to understand):
"We then fixed the base model and fine-tune the
fully connected layer to predict replication timing of genomic loci in the GM12878 cell line. Next, with the
attention network fixed, we fine-tuned the base model with the task of"

- The adjacency matrix $A=H.W.H^T$ corresponds to a graph where any two nodes connected by an hyperedge are now connected by an edge. Correct ?
This is quite a reduction of the hypergraph.
Maybe a small comment about this would be helpful for the reader.

- sec 3.1 : you could insert somewhere, at beginning of a paragraph "Here we recall the basic steps of SSL applied to our hypergraph problem, for completeness". I say this because there is basically no new information in that part, for a reader who knows the basic working of SSL.
Yet, I found it's well written and pedagogical: I would keep it, but put this disclaimer at the beginning, to save time for most readers.

- "By integrating the model architecture
and pre-training scheme, PhyGCN effectively learns from a hypergraph structure and applies the learned
knowledge to downstream tasks."
-> useless, informative-less sentence. I suggest to delete.

- "Based on the pre-trained model f (·) that generates node embeddings with the information
it leveraged from the pre-training task, we can utilize it to do node-level downstream tasks in a wide range
of applications."
-> again, useless sentence.

- page 6, top, typo $E'_valid$ and $E'_valid$

- I infer from the equation of page 6 (top) that z_i is the binary variable that identifies the positive and negative samples (hyperedges belonging to the positive or negative set, for a given hypergraph).
It would be easier to write it down.

- continual or continuous ? Define continual.

- by the red arrow
->
by the blue arrow

- typo, titles of fig 5a, 5b:
AUROC score on different datasets
->
AUROC score with different models

**Strengths And Weaknesses:**

Strengths:
- The paper is overall well written, except from a few points (see below)
- Experiments are sound. (I have a few important demands, see below).
- Methodology is serious too (e.g. using the same train/val splits to compare various methods)

Weaknesses
- The paper novelty is quite limited, but credit is given appropriately, so it seems fine (I didn't search to check if other works applied SA-GNN as a pretraining step for hypergraphs related tasks).
- Maybe hypergraphs is a niche subject ? But it sounds general enough to be relevant to TMLR.
- some sentences are useless and carry no information, are tiring to read. Please remove them (see below)


Comment:
- Skip/Dense connection (to avoid vanishing gradient): What about a ResNet structure ? Did you try that ?
Your architecture is nice in terms of less resources used, but then, you are demanding the network to produce intermediate representations that are useful for the (pretext) task, instead of allowing intermediate steps to be hard-to-decode, but maybe useful to build high-order representations.
Anyway, that's your choice.

---

> ### Author Response · Authors · 2024-06-13
> **Response to the reviewer's comments (part one)**
>
> Thank you very much for your helpful comments and suggestions. We answer your questions as follows. We have also revised our paper accordingly to incorporate your suggested changes.
>
> ---
> **Q:** In fig 3, why is SAGNN (Zhang 2020) not presented ?
>
> **A:** We note that HyperSAGNN is a method designed for hyperedge prediction, and is therefore only considered as a baseline on hyperedge prediction tasks. We had included a comprehensive discussion on HyperSAGNN and PhyGCN (encoder vs. hypergraph convolution) on hyperedge prediction as in Figure 5a/b in our paper and Tables 6, 8, 9 and 10 in the appendix.
>
> For node classification tasks in our Figure 3, comparing with HyperSAGNN is not possible. Nevertheless, we add the ablation study on changing our base hypergraph convolutional network to one that is used in HyperSAGNN (i.e., convolution v.s. encoder). The results are updated in Table 4 in the appendix, which shows the better performance of a convolution-based network.
>
> ---
> **Q:** In the experiment section, the size of the pre-training dataset is not mentioned. For instance, when training with 0.5% of the available data, how is the pre-training done? With the same amount of raw hypergraphs, but using numerous randomized samplings to self-supervise in many ways ? Or, was a larger corpus of hypergraphs used ? This is a very important point, that must be mentionned explicitly, regardless of the choice you made.
>
> **A:** We thank the reviewer for this comment and apologize for the potential confusion. The 0.5% of available data only refers to the available label class for the nodes, which is the training data at the finetuning or node classification evaluation stage. The pretraining is done on all available hyperedges with negative samplings. The experiments are designed under the assumption that labeled node class labels are less abundant than observed hyperedges.
>
> ---
> **Q:** Why limit to 4% of the data at maximum?
>
> **A:** We thank the reviewer for the suggestion and now add 2 more splits of 8% and 16% as in our updated Figure 2b, where PhyGCN consistently outperforms its baselines on the new data ratio. Note that in a lot of previous deep learning methods using these datasets as benchmarking, the training data is set to be 1% as a constant (HyperGCN, HNHN, etc.).
>
> ---
> **Q:** The pre-training loss is not written down in the main text.
>
> **A:** The pre-training loss is the Binary Cross Entropy (BCE) loss between models’ prediction and the ground truth of whether (hyper)edges exist. We presented the detailed formulation of the loss in Eq. (2) of section 3.3 on page 6.
>
> ---
> **Q:** The outcome of the experiment presented in fig 5 is insufficiently commented on.
>
> **A:** We previously discussed the results of Figure 5(a) (b) and (c) separately in the main text of Section 4.3. We have added more comments and discussion in our revision.

---

> ### Author Response · Authors · 2024-06-13
> **Response to the reviewer's comments (part two)**
>
> ---
> **Q:** Why PhyGCN is not affected by the affected/unaffected split?
>
> **A:** For the expected and unexpected data splits in polypharmacy side effect prediction, expected splits refer to cases where a hyperedge already contains an edge, making it easier to predict even if the model has only learned low-level information from a binary graph structure. In contrast, unexpected splits require models to thoroughly learn and understand the hypergraph structure to make accurate predictions. The robustness of PhyGCN on both expected and unexpected splits indicates that PhyGCN effectively learns and leverages the hypergraph structure.
>
> ---
> **Q:** What do you mean by variable-sized hyperedge prediction?
>
> **A:** By variable-sized hyperedge prediction, we mean that the size of the hyperedge may vary (e.g., size k could take the value of 2, 3, or 4). On the contrary, fixed-sized hyperedge prediction refers to predicting with hyperedges that only have one fixed size. We have added this explanation to our revision on page 2.
>
> ---
> **Q:** What are the two types of negative sampling strategies?
>
> **A:** For the task of predicting the side effects of a drug combination, i.e., the hyperedge among (drug1, drug2, side effect 1), the most commonly used negative sampling strategy (referred to as the original negative sampling) would sample two random drugs with a given side effect. These sampled drugs mostly do not cause any side effect, simplifying the prediction so that a model which can predict if a drug pair causes ANY side effect will have satisfactory performance. Our enhanced negative sampling strategy, instead of sampling random drug entities, selects an incorrect pair of drugs known to cause other side effects and combines it with the given side effect that it does not cause, therefore increasing the prediction difficulty. .
>
> We reason this negative sampling strategy resembles a harder task as supported by lower AUROC when evaluating performance. The task for predicting drug pair side effects can actually be seen as a two-step task: 1. A binary classification task, of whether a drug pair can cause any side effect or not; 2. A multi-class classification task, of what specific side effects a drug pair can cause if it will cause side effects. The first task due to the inherent imbalance in the dataset can result in much higher AUROC scores for all methods, and the original negative sampling strategy would be biased towards that. The second task is harder and is what our enhanced negative sampling strategy is trying to highlight.
>
> ---
> **Q:** AUROC and AUPR are acronyms and are not defined
>
> **A:** Thank you for catching it. We now define them in the main text.
>
> ---
> **Q:** Whether fine-tuning means training the decoder (h) weights only, or also those of the base model f
>
> **A:** Both the base model f and the decoder are fine-tuned.
>
> ---
> **Q:** Other writing suggestions.
>
> **A:** Thank you for these helpful writing suggestions. We have incorporated them all into the revision.

---

### Review · Reviewer_scvy · 2024-05-28

**Summary Of Contributions:**

This study presents a new hypergraph convolutional neural networks as well as a pre-training and self-supervised learning framework, called PhyGCN. PhyGCN is based on a self- supervised task that predicts masked hyperedges from observed ones. To this end, it uses an attention module to predict variable-sized hyperedges.

**Audience:**

Yes

**Claims And Evidence:**

No

**Requested Changes:**

1. Detailed discussion of **recent** related work.
2. Clearly mention the main messages of this paper and discuss how this method can address existing limitations of **recent** studies.
3. Clearly mention that which of the experiments are designed to support the above messages.

**Strengths And Weaknesses:**

Despite the facts that the paper is well-written, easy to follow, has discussed an important problem, and is also interesting for at least some individuals in TMLR's audience, there are some unclear points that require more discussion:

1. The paper lacks clear research questions: Generally, the paper is not well-motivated and requires more discussion on the limitations of existing methods. What are the main messages of this study? Why do we need PhyGCN, and what does it have that existing methods don’t? I realized that the authors several times discussed the importance of predicting variable-sized hyperedges.  While it had been a serious challenge back in 2021-2022, several existing methods are capable of this.

2. There is a lack of discussion on recent studies! This paper has been submitted in 2024, and there is no discussion on even a single paper from 2023. This lack of discussion on recent studies has caused several unclear points about the main messages of this paper. For example, several recent methods are capable of self-supervised training and also can be generalized to unseen data, see [1, 2, 3]. These studies also are capable of handling hyperedges with variable sizes. Other claims are also discussed in several studies. For example, [1] also theoretically discussed the fact that using simple averaging of nodes’ encodings is not enough for hyperedge prediction tasks.

3. As discussed above, since there are no clear research questions in the paper, it is hard to evaluate if the experiments support the claims or not.


---
[1] CAT-Walk: Inductive Hypergraph Learning via Set Walks. Behrouz et al. NeurIPS 2023.
[2]  Hypergraph neural networks through the lens of message passing: a common perspective to homophily and architecture design. Telyatnikov et al. 2023.
[3] I’m me, we’re us, and i’m us: Tri-directional contrastive learning on hypergraphs. Lee and Shin. AAAI 2023.

---

> ### Comment · Reviewer_d2MZ · 2024-06-03
> **Approval of main comment**
>
> I agree with reviewer scvy that recent literature should be better discussed and compared with.
> This is a crucial point.

---

> ### Author Response · Authors · 2024-06-13
> **Response to the reviewer's comments**
>
> Thank you very much for your helpful comments and suggestions. We answer your questions as follows. We have also revised our paper accordingly to incorporate your suggested changes.
>
> ---
> **Q:** Request a detailed discussion of recent related work.
>
> **A:** Thank you for pointing out these related papers. We have added the discussion of them in the revised introduction section. In particular, [1] studies temporal hypergraphs and uses random walk to automatically extracts temporal, higher-order motifs. [2] studies how the concept of homophily can be characterized in hypergraphs, and proposes a novel definition of homophily to effectively describe HNN model performances. Both works are related to our study on hypergraphs but orthogonal to our focus on the performance improvement by self-supervised learning on hypergraphs. [3] studies self-supervised learning on hypergraphs and utilizes a tri-directional contrastive loss function consisting of node-, hyperedge-, and membership-level contrast. Very recently [4] studies generative self-supervised learning on hypergraphs and proposed a new task called hyperedge filling for hypergraph representation learning. Note that [2] and [4] are released after our work appeared online.
>
> We added experiments to compare our method PhyGCN with the most relevant methods proposed in [3] (Tri-CL) and and [4] (HypeBoy) as additional baselines for performance comparison in Table 3 of Appendix B.2. Our results indicate that all three methods perform comparably, with each method excelling on different datasets. PhyGCN generally ranks as either the best or second-best method across various datasets and dataset splits. Notably, HypeBoy necessitates extensive softmax computation for nodes against all other nodes, resulting in significant GPU memory consumption and limited scalability to large datasets. Conversely, PhyGCN’s self-supervised learning framework is lightweight and scalable, requiring minimal computational resources. TriCL, on the other hand, involves much more hyperparameters that need careful tuning for each dataset due to its tri-directional contrastive loss function. In contrast, PhyGCN only requires tuning the number of layers for different datasets.
>
> References:
> [1] CAT-Walk: Inductive Hypergraph Learning via Set Walks. Behrouz et al. NeurIPS 2023.
> [2] Hypergraph neural networks through the lens of message passing: a common perspective to homophily and architecture design. Telyatnikov et al. 2023.
> [3] I’m me, we’re us, and i’m us: Tri-directional contrastive learning on hypergraphs. Lee and Shin. AAAI 2023.
> [4] HypeBoy: Generative Self-Supervised Representation Learning on Hypergraphs. Kim et al. ICLR 2024.
>
> ---
> **Q:** The research question and the main message of this paper.
>
> **A:** The research questions that we aim to answer are: How can we enhance node representation learning in hypergraphs by leveraging self-supervised learning methods to effectively utilize abundant unlabeled data? How can we fully capture the hypergraph structure? We briefly illustrate our main message as follows. This paper proposes a novel method that leverages the hypergraph structure for self-supervision to enhance node representation learning. By introducing a unique training strategy capable of adapting to variable hyperedge sizes with self-supervised learning, PhyGCN improves generalization to unseen data. Methods that can only handle fixed hyperedge sizes lack the flexibility to train on diverse data structures. The effectiveness of PhyGCN is demonstrated through applications on multi-way chromatin interactions and polypharmacy side-effects, showing its potential for enhancing hypergraph node representations across various domains.
>
> ---
> **Q:** Which of the experiments are designed to support the above messages.
>
> **A:** We presented several different experiment settings. Specifically, we performed:
>
> 1. **Node Classification Tasks:** Evaluations on node classification tasks included citation networks using different training data ratios. We demonstrated that PhyGCN consistently outperforms baselines across various data splits, highlighting the model's robustness, its ability to learn from higher-order hypergraph structures, and its effective utilization of the hypergraph structure. The superior performance of PhyGCN compared to GCN demonstrates that PhyGCN captures hypergraph structure beyond merely decomposing it into graph structure.
>
> 2. **Downstream Applications on Real-world Problems:** Evaluations included multi-way chromatin interaction prediction tasks and polypharmacy side effect prediction. On these more complex real-world problems, PhyGCN outperforms MATCHA in predicting multi-way chromatin interactions, demonstrating the effectiveness of the hypergraph convolutional architecture. Moreover, PhyGCN performs consistently better on both expected and unexpected data splits in polypharmacy side effect datasets, indicating its superior ability to model complex interactions as hypergraphs.

---

### Author Response · Authors · 2024-06-13
**General response and summary of revisions**

Dear Editor and Reviewers,

Thank you for your time in providing valuable feedback on our manuscript. We have incorporated all your suggestions in our revision, with the revised texts highlighted in blue. To summarize, we made the following revisions to improve the presentation, the comparison with the literature, and the discussion of our contribution:

1. We have added a subsection in the introduction called "Most Related Work" to clearly distinguish our work from recent studies on self-supervised pre-training of hypergraphs.

2. We have added new experiments to compare with recently published methods (HypeBoy and Tri-CL). The experimental results further confirm our main message in the paper that self-supervised learning on hypergraph structures is a meaningful task, and our method provides a more scalable way to achieve it.

Best,

Authors

---

### Comment · Reviewer_d2MZ · 2024-06-21
**Answer to first rebuttal**

Dear Authors,

First, please note that TMLR does not have a length limit, as sepcified in https://jmlr.org/tmlr/author-guide.html :
> Submissions may be any length, but a paper’s length should be justified by its content and papers that are unusually long (not counting any Appendices) are likely to result in reviewing delays.

I thank you for your detailed answers.

Remarks:
- table 3 and related appendix text should be moved up into the main text, since this is a very relevant comparison with recent, similar works.
- you assume prior knowledge of the general idea about negative sampling and the role of negative samples in SSL. A couple of sentences about this, early on, would help the unfamiliar reader (especially that your approach then differs from traditional transfer learning, see below). Please use precise words or maths rather than general words to explain it. You could for instance write the (SSL) loss in a simplified way, for a generic SSL setup with positive/negative samples (just to give the idea).

Overall, your answers to our questions are quite complete, but the changes in the paper do not reflect that. For instance, to my question " What are the two types of negative sampling strategies?", your answer was very clear, but was not included in the text. I would really advocate for clear wording (as in your answers, avoiding long & technical sentences), letting some space for intuitive reasoning (in the manuscript), for the sake of clarity.

So, I am asking that you further improve the manuscript, in particular on the following points:
- literature survey: since there is no length limit, take the time to describe each paper contribution (and how it relates/differs from yours when needed). Or explicit why you think it is irrelevant to your research question.
- As requested by reviewer ySwi, "Add more details on the used datasets.", I suggest to explicit the tasks one by one : input type (shape, nature, etc), output type (shape of data, nature, number of labels, etc). Be as explicit as possible, it will only help the reader's understanding. Provide the figures about datasets (number of hypergraphs, typical number of nodes or number of hyperedges, possible sizes of hyperedges, etc.)
- following your answer to the question "Add more explanations about the experiment "Transferring knowledge to a different hypergraph structure", you answer "We apologize for the confusion. The reviewer is correct that *our problem setting is unique and different from the traditional inductive learning setting*....[detailed explanation]". This is **very** important and it **must** appear very clearly in the paper.
- figure 5 & the new pre-training strategy: in the text, it's still not very explicit at first sight, what can be deduced from the experiment means. You could be more explicit in the text, using simple words, as you did in your answer.
- loss function: could go with more explanations. Currently, it's pretty hard to understand. I think it's related to the fact you use the term "samples" to refer to the hyperedges of a given hypergraph G=(V,E). In Eq. (2), the Edges appear, but for a single graph G ? I guess in practice you sum or average over many (hyper) graphs. By default, in a GNN setup, I call a "sample"  each graph in the training set, not each edge. I think this is the origin of the confusion.
- downstream task: provide an example of loss function (I mean, give the loss in at least one application case)

Wording suggestions, for clarity:
- fine tuning of base model/last layer: please be explicit in your wording, e.g. page 3, "subsequently fine-tune it with the fully connected layer for downstream tasks." -> "subsequently fine-tune the base model together with the fully connected layer for downstream tasks." (ok I see it's written below Eq 4, but that's way later)
- Suggestion for wording: keep recalling the role of each object in terms of nodes/edges  when explaining your ideas. For instance,
    - "randomly sampling two drugs that do not cause the specific side effect in the training data"
    - would become:
    - "randomly sampling two drugs (nodes) that do not cause the specific side effect (node) in the training data"

Maybe it's just me, but:
- difference with SAGNN (Zhang 2020): should be outlined in the paper. Currently it seems like you share the same backbone, when reading the paper. It's only at the appendix, We evaluated our base convolutional model against the encoder-decoder model used in Tu et al. (2018); Zhang et al. (2020) for representation learning, using the same attention network from Zhang et al. (2020) for combining the node embeddings and making the prediction." that the reader can understand, implicitly, that your model is actually not the same as Zhang's (having not read Zhang et al, that is).

typos:
- figure 5: red arrow: isn't this the blue arrow ?

---

> ### Author Response · Authors · 2024-06-23
> **Thank you for your feedback and suggestions**
>
> Thank you for your further feedback and suggestions. We will revise our paper again accordingly.
>
> Authors.

---

> > ### Author Response · Authors · 2024-06-25
> >
> > We thank the reviewer for your further detailed feedback on the wording and writing of our manuscript. We have revised all points you raised accordingly in the new revision. We thank you again for making the manuscript more polished.

---

> > > ### Comment · Reviewer_d2MZ · 2024-06-26
> > > **thank you**
> > >
> > > Thank you, I see you have improved the manuscript in many ways, I now let the othere reviewers make their comments.
> > >
> > > In particular, your original negative sampling strategy is now very clearly presented, thank you.

---

### Decision · Action_Editor_ZK76 · 2024-07-12

**Recommendation:** Accept as is

**Comment:**

All reviewers acknowledge that while this work may lack novelty, it presents a nice combination of existing ideas, offering statistical distinctions from current hypergraph pre-training efforts. The research questions tackled are significant for a variety of hypergraph-related downstream tasks. The authors have effectively demonstrated use cases and provided comprehensive dataset descriptions. All claims have been substantiated, and the distinctions between this work and existing literature are well delineated after revision. Therefore, I recommend acceptance.

**Audience:**

Although hypergraphs are a niche subject, they are sufficiently general to be relevant to a specific audience within TMLR.

**Claims And Evidence:**

This manuscript introduces a graph neural network pretraining method tailored for hypergraphs. Key contributions include the design of a pre-training task—hyperedge prediction—which incorporates a negative sampling strategy for semi-supervised learning (SSL) on the original hypergraph-capable GNN, Hyper-SAGNN (Zhang 2020). All reviewers unanimously concurred that the claims have been thoroughly substantiated following the rebuttal and revision process.